# *Haberlea rhodopensis* Extract Tunes the Cellular Response to Stress by Modulating DNA Damage, Redox Components, and Gene Expression

**DOI:** 10.3390/ijms242115964

**Published:** 2023-11-04

**Authors:** Dessislava Staneva, Neli Dimitrova, Borislav Popov, Albena Alexandrova, Milena Georgieva, George Miloshev

**Affiliations:** 1Laboratory of Molecular Genetics, Epigenetics and Longevity, Institute of Molecular Biology “Roumen Tsanev”, Bulgarian Academy of Sciences, 1113 Sofia, Bulgaria; dessysta@gmail.com (D.S.); milenageorgy@gmail.com (M.G.); 2Department of Molecular Biology, Immunology and Medical Genetics, Faculty of Medicine, Trakia University, 6000 Stara Zagora, Bulgaria; dimitrovanelly83@gmail.com (N.D.); dr_b_popov@abv.bg (B.P.); 3Laboratory of Free Radical Processes, Institute of Neurobiology, Bulgarian Academy of Sciences, 1113 Sofia, Bulgaria; aalexandrova@abv.bg

**Keywords:** *Haberlea rhodopensis*, gamma irradiation, oxidative stress, genotoxicity, antioxidant, gene transcription, comet assay, flow cytometry

## Abstract

Ionizing radiation (IR) and reactive oxygen species (ROS)-induced oxidative stress can cause damage to cellular biomolecules, including DNA, proteins, and lipids. These harmful effects can compromise essential cellular functions and significantly raise the risk of metabolic dysfunction, accumulation of harmful mutations, genome instability, cancer, accelerated cellular senescence, and even death. Here, we present an investigation of HeLa cancer cells’ early response to gamma IR (γ-IR) and oxidative stress after preincubation of the cells with natural extracts of the resurrection plant *Haberlea rhodopensis*. In light of the superior protection offered by plant extracts against radiation and oxidative stress, we investigated the cellular defence mechanisms involved in such protection. Specifically, we sought to evaluate the molecular effects of *H. rhodopensis* extract (HRE) on cells subjected to genotoxic stress by examining the components of the redox pathway and quantifying the transcription levels of several critical genes associated with DNA repair, cell cycle regulation, and apoptosis. The influence of HRE on genome integrity and the cell cycle was also studied via comet assay and flow cytometry. Our findings demonstrate that HREs can effectively modulate the cellular response to genotoxic and oxidative stress within the first two hours following exposure, thereby reducing the severity of such stress. Furthermore, we observed the specificity of genoprotective HRE doses depending on the source of the applied genotoxic stress.

## 1. Introduction

Extensive studies of medicinal plants and their phytochemical compounds attract scientific attention [1] due to their proven radioprotective [2,3], antioxidant, and immunomodulating potential [3,4,5,6], as well as their anticancer activity, which is very often executed by arresting dividing cells during the cell cycle [7].

The resurrection plant, *Haberlea rhodopensis* (Friv.), the Orpheus flower, is a rare Balkan endemic plant. It is the first species from the *Gesneriaceae* family discovered in the Balkans [8]. It is rarely distributed in several specific regions, like the Middle and Eastern Rhodope Mountains, the Middle Balkan Mountains, and the Predbalkan Mountains in Bulgaria [9]. It is proven that the plant *H. rhodopensis* experiences serious genetic reprogramming under severe drought, directing resources from growth and development to cell protection [10]. *H. rhodopensis* could survive after prolonged desiccation; therefore, it was named the resurrected plant.

The chemical composition of *H. rhodopensis* was studied by many groups and allowed the discovery of more than one hundred bioactive compounds, including fatty acids and sterols, saccharides, flavonoids, tannins, and polysaccharides [6,11,12,13,14,15,16]. Among the main compounds isolated from *H. rhodopensis* were myconoside, paucifloside, and three new flavone *C*-glycosides, i.e., hispidulin 8-*C*-(2-*O*-syringoyl-β-glucopyranoside), hispidulin 8-*C*-(6-*O*-acetyl-β-glucopyranoside), and hispidulin 8-*C*-(6-*O*-acetyl-2-*O*-syringoyl-β-glucopyranoside) [17,18]. In addition, the potent antioxidant and hepatoprotective effects of myconoside were recently evidenced [3,19].

Generally, phenolic acids accumulated in high amounts in the resurrection plants possess therapeutic properties due to their ability to capture free radicals and decrease oxidative stress [6,19,20,21,22]. For example, Berkov and colleagues studied by gas chromatography-mass spectrometry the polar and apolar fractions of methanol *H. rhodopensis* extracts (HRE) and identified five free phenolic acids, namely syringic, vanillic, caffeic, dihydrocaffeic, and p-coumaric [23]. Furthermore, another study demonstrated that in alcohol HRE, the most abundant phenolic acids were the sinapic, ferulic, caffeic, and p-coumaric acids, as well as at least five other phenolic acids, which, although not so abundant, were still present in the extracts [24].

Notably, alcohol extracts from *H. rhodopensis* were shown to possess antioxidant, antiviral, antibacterial, and antifungal activities [6,16,25,26,27,28,29]. By assessing the reduced number of abnormal cells and chromosomal aberrations in gamma-irradiated (γ-IR) rabbit lymphocytes treated with HRE, other authors and we have evaluated the in vivo radioprotective (γ-IR), anticlastogenic, and antimutagenic potential of the extracts against the carcinogen cyclophosphamide [17,22,30,31,32,33]. In addition, pretreatment with HRE significantly elevated the activity of specific antioxidant enzymes like superoxide dismutase (SOD) and catalase (CAT). At the same time, it had an anti-lipid peroxidative effect by reducing plasma malondialdehyde (MDA) levels in blood plasma [28,34,35]. Furthermore, along with the radioprotective effects, HRE showed in vivo immune-stimulatory, anti-tumour, and anti-inflammatory activities [30,36,37,38,39]. Moreover, using another model system, the yeast *Saccharomyces cerevisiae*, we have proven that methanol HRE possesses anti-ageing activity [40]. It should be noted, however, that regardless of the detailed chemical analyses of *H. rhodopensis* extracts, the profound molecular mechanisms of their cell protective effects remain to be elucidated.

In this study, we report on the early response of mammalian cells (specifically, human cervix epithelial carcinoma, or HeLa cells) to genotoxic stress following pretreatment with varying concentrations of ethanol HRE and subsequent exposure to γ-IR, or oxidative stress. Our results demonstrate the genoprotective properties of HRE, which can be attributed to its modulatory effect on intrinsic players within cellular redox systems, cell cycle regulation, and the expression of genes involved in DNA damage and repair pathways. Moreover, we provide unequivocal evidence that, within the first several hours following irradiation and oxidative stress, *H. rhodopensis* extracts effectively mitigate the severity of the genotoxic response.

## 2. Results

### 2.1. Myconoside Is the Dominant Compound in the Prepared Total Ethanolic Haberlea rhodopensis Leaves Extract

Studies on *Haberlea rhodopensis* have identified the caffeoyl phenylethanoid glycoside myconoside [β-(3,4-dihydroxyphenyl)-ethyl-3,6-di-O-β-D-apifuranosyl-4-O-α,β-dihydrocaffeoyl-O-β-D-glucopyranoside] as one of its main biologically active ingredients [17,18,38,39]. Considered the predominant active compound in *Haberlea rhodopensis* extracts, our study focused on identifying and quantifying myconoside in the prepared total ethanol HRE using high-performance liquid chromatography (HPLC) analysis. The most abundant peak in the chromatogram corresponded to myconoside (Figure 1). The HPLC retention time of myconoside in the extract was seven minutes, and its content was determined to be 140 ± 2.3 mg/g dry weight.

### 2.2. No Impact of HRE on Cellular Morphology and Cell Cycle Progression in HeLa Cells

Flow cytometry analysis was employed to investigate the impact of HRE treatment and H_2_O_2_ and γ-IR genotoxins on cellular characteristics, including size (forward scatter, FSC), granularity (side scatter, SSC), and cell cycle progression (DNA content) in the initial hours following the treatment. The comparison of treated and control samples revealed no substantial changes in cellular morphology, as observed in Appendix A. Furthermore, the distribution of cell populations across different cell cycle phases showed no significant variations regardless of the treatment, as depicted in Appendix A.

### 2.3. Exploring the Protective Potential of HRE Pretreatment against Genotoxicity in HeLa Cells

HeLa cells were subjected to pretreatment using various concentrations of HRE, followed by exposure to genotoxic stress induced by H_2_O_2_ or γ-IR, as outlined in the Materials and Methods section. Furthermore, the alkaline comet assay (CA) was employed to assess the level of genotoxicity induced in the cells. The resulting outcomes, presented in Figure 2, illustrate the average values of the Olive moment (OM) parameter derived from triplicate experiments.

Notably, incubation with HRE at 10 and 25 μg/mL did not induce cell genotoxicity. Only the highest tested concentration of 50 μg/mL HRE caused a statistically significant increase in the Olive moment critical parameter (Figure 2, blue bars). However, considering the apparent increase in OM provoked by the stressors, the 17% rise detected for 50 μg/mL HRE could be considered a common genotoxic effect. Conversely, the two genotoxic stressors (H_2_O_2_ and γ-IR) caused DNA damage. The OM values were significantly higher in genotoxin-treated cells than untreated control cells, exhibiting an average increase of 2.66-fold for H_2_O_2_-treated cells and 1.73-fold for γ-IR-treated cells. Intriguingly, pretreatment with HRE conferred protection against genotoxic damage in the cells.

Interestingly, an escalating gradient of HRE concentrations during pretreatment augmented the safeguarding effect against H_2_O_2_-induced DNA damage, whereas the opposite trend was observed for genotoxicity induced by γ-IR. Specifically, as the HRE concentration increased, the protection against 2 Gy γ-radiation decreased (Figure 2, last three bars). These notable discrepancies in the genoprotective capacity of HRE prompted a comprehensive investigation into the underlying molecular mechanisms of HRE action.

### 2.4. Exploring the Beneficial Effects of Pretreatment with HRE on Cellular Antioxidant Status: Protection against H_2_O_2_ and γ-IR-Induced Stress

To assess the oxidative stress status, the activities of SOD, CAT, and glutathione peroxidase (GPx) enzymes, as well as the levels of total glutathione (tGSH) and lipid peroxidation (LPO), were quantified in all four experimental groups: control cells (HRE^−^ H_2_O_2_^−^/γ-IR^−^), cells treated with HRE (HRE^+^ H_2_O_2_^−^/γ-IR^−^), cells exposed to genotoxic stress (HRE^−^ H_2_O_2_^+^/γ-IR^+^), and cells pre-incubated with HRE before oxidative (10 mM H_2_O_2_) or γ-IR (2 Gy) stress (HRE^+^ H_2_O_2_^+^/γ-IR^+^). The obtained results are summarised in Figure 3.

#### 2.4.1. Modulation of Superoxide Dismutase Activity by HRE and Genotoxic Stress

Treatment of cell cultures with various concentrations of HRE resulted in a significant average induction of SOD activity by 38 ± 3.16%, independent of HRE concentration (Figure 3A). At their respective doses, both genotoxic agents, H_2_O_2_ and γ-IR, caused an approximate 50% increase in SOD activity (100% vs. 150.5% and 156%, respectively) (Figure 3B,C). Preincubation of cells with HRE before oxidative stress reduced SOD activity to 134 ± 3.09%, comparable to the level observed in the HRE^+^ H_2_O_2_^−^ group. Notably, concentrations of HRE ranging from 25 to 50 μg/mL exhibited the most pronounced effects (*p* < 0.05). Conversely, irradiation of HRE-pretreated HeLa cell cultures showed a slight cumulative effect, dependent on the dose. Comparison between HRE^+^ γ-IR^+^ and HRE^−^ γ-IR^+^ groups revealed a modest decrease in SOD activity at 10 μg/mL HRE and a 15% increase at 50 μg/mL HRE.

#### 2.4.2. Impact of HRE Administration, Oxidative and Radiation Stress on Catalase Activity

As shown in Figure 3A, administration of HRE to the cells (at any of the concentrations tested) did not cause a statistically significant alteration in the activity of the CAT enzyme. After treatment with hydrogen peroxide, catalase activity increased by nearly 30% (*p* = 0.004), but it gradually decreased as the amount of applied HRE increased (Figure 3B). When cells were preincubated with 50 μg/mL of the extract, the CAT activity returned to the control group value. The CAT activity was 15% higher in γ-rays-irradiated cells compared to non-irradiated ones. A decrease was observed when cells were subjected to the combined treatment of HRE plus radiation (Figure 3C).

#### 2.4.3. Modulation of Glutathione Levels by HRE in Response to Oxidative and Radiation Stress

Figure 3A demonstrates that incubation of HeLa cells with different concentrations of HRE did not elicit any significant changes in the levels of total glutathione (tGSH) compared to the untreated control group. However, exposure to oxidative stress induced by 10 mM H_2_O_2_ resulted in a twofold decrease in tGSH levels compared to the non-treated control (HRE^−^ H_2_O_2_^+^ vs. HRE^−^ H_2_O_2_^−^, *p* < 0.05). Notably, preincubation with HRE preserved glutathione content closer to normal levels (*p* < 0.05 vs. HRE^−^ H_2_O_2_^+^) (Figure 3B). The most effective protective effect was observed with 10 and 25 μg/mL of the HRE extract, where tGSH levels reached 84% of the control value (*p* > 0.1 vs. HRE^−^ H_2_O_2_^−^). Following exposure to 2 Gy radiation, a slight decrease (less than 10%) in tGSH levels was observed during the initial hours (Figure 3C). Interestingly, in irradiated samples preincubated with ethanolic *Haberlea* extracts, a noticeable, concentration-dependent increase in tGSH levels was detected (2.4 times higher than in HRE^−^ γ-IR^+^). The combined treatment of γ-IR with HRE demonstrated a potentially synergistic effect, resulting in an elevation of tGSH levels at higher concentrations of HRE. Overall, the results indicate a decrease in tGSH content in cells subjected to either H_2_O_2_ or γ-IR stress, while preincubation with HRE significantly increased glutathione levels.

#### 2.4.4. Modulation of Glutathione Peroxidase Activity by HRE in Response to Oxidative and Radiation Stress

At concentrations above 10 μg/mL, HRE caused a slight reduction in GPx activity, although these changes did not reach statistical significance (*p* > 0.1; Figure 3A). Treatment with H_2_O_2_ decreased GPx activity, and preincubation with the *Haberlea* extract further diminished the enzyme activity (*p* < 0.05; Figure 3B). Notably, exposure to 2 Gy radiation inflicted a 30% reduction in GPx activity, whereas pretreatment with 10 μg/mL HRE led to a 17% increase in enzyme activity (70.8% vs. 83.3%). Concentrations of HRE above 10 μg/mL exhibited a further decline in GPx activity (Figure 3C).

#### 2.4.5. Modulation of Lipid Peroxidation and Antioxidant Defense by HRE in HeLa Cells

Figure 3A demonstrated that all tested HRE concentrations elevated the lipid peroxidation level by more than twofold in HeLa cells. Expectedly, the two stressors, H_2_O_2_ and 2 Gy IR, increased LPO levels by 3.1 and 1.44 times, respectively (Figure 3B,C, red bars). Interestingly, 25 μg/mL of HRE exhibited the highest efficacy in neutralising lipoperoxides, resulting in a 33% decrease compared to H_2_O_2_-treated cells (Figure 3B, LPO). However, even at this concentration, the LPO level remained two times higher than that of the control group (HRE^−^ H_2_O_2_^−^, green line in Figure 3B). Similarly, cells exposed to γ-IR displayed a 43% increase in LPO (Figure 3C) compared to untreated cells. Combining higher concentrations of HRE pretreatment with gamma irradiation further augmented lipid peroxides, albeit concentration-dependent. Overall, the extract induced lipid peroxidation in cells, and after combined treatment with HRE and H_2_O_2_ or γ-IR, LPO levels remained higher than those in the control group. It should be noted that the observed increase in MDA levels, a marker of lipid oxidation, shortly after treatment may be influenced by other substances present in *Haberlea* extracts, such as polysaccharides, which can interact with the assay reagent used for MDA measurement [41,42]. Thus, the observed phenomenon could result from the incredible natural abundance of polysaccharides in the *Haberlea* extracts, which cross-react with TBA. In addition, others have demonstrated that flavonoids and polyphenols, enriched in *Haberlea* extracts, may exhibit dual activities and act as both antioxidants and prooxidants, issuing opposite outcomes [43,44,45].

In conclusion, the HRE demonstrated the ability to induce lipid peroxidation while increasing SOD activity, thereby priming the cells to cope with subsequent stressors.

### 2.5. Influence of H. rhodopensis Extract on Gene Expression in Response to Oxidative Stress and Radiation

The observed effects of HRE on critical enzymes of the cellular redox system highlight its influence on cellular responses. However, the cellular stress response is a complex process involving various systems, genes, and networks. To better understand how HRE affects the cellular response to oxidative stress and radiation, we investigated the expression of critical genes involved in the cell cycle, DNA repair, and signalling. Specifically, we examined the transcription levels of ATM, BRCA1, CDKN1A (p21), RAD50, RAD51, and BBC3 (PUMA) in the investigated experimental groups, including control cells (HRE^−^ H_2_O_2_^−^/γ-IR^−^), cells treated with HRE alone (HRE^+^ H_2_O_2_^−^/γ-IR^−^), and cells pretreated with HRE and subsequently exposed to oxidative stress (10 mM H_2_O_2_) or ionizing radiation (2 Gy γ-IR) (HRE^+^ H_2_O_2_^+^/γ-IR^+^). The transcript levels were evaluated two hours after treatment to elucidate the early molecular mechanisms underlying the cellular response to stress, such as identifying the genes that exhibit the earliest changes in expression within the stress response signalling cascade. The outcomes of these gene expression analyses are presented in Figure 4.

#### 2.5.1. Evaluation of ATM Gene Expression

The impact of HRE on the expression of the ATM gene, a critical component of the cellular stress response, was investigated in HeLa cells. The addition of HRE to the cells (Figure 4A) and incubation with H_2_O_2_, with or without prior treatment with the extract (Figure 4B), did not result in significant changes in ATM transcript levels within the examined period. However, exposure to 2 Gy radiation induced a non-significant increase of approximately 50% in ATM mRNA expression (Figure 4C). Interestingly, incubation of cells with 10 μg/mL HRE before irradiation maintained ATM gene expression at the elevated level observed in irradiated cells. In contrast, pretreatment with 25 μg/mL HRE significantly decreased ATM gene expression by approximately two-fold compared to the HRE^−^ γ-IR^+^ group, approaching the level seen in the control group (*p* < 0.05).

#### 2.5.2. Influence of *H. rhodopensis* Extract on RAD50 Gene Expression in Response to Oxidative Stress and Radiation

The effect of HRE on the transcription of the RAD50 gene, another key player in cellular stress response, was examined in HeLa cells. In the initial hours following treatment, modest yet statistically significant changes in RAD50 gene expression were observed in cells treated with HRE alone. At a concentration of 10 μg/mL, RAD50 expression was upregulated by 15%, while at 50 μg/mL, it was downregulated by 10% compared to the non-treated control group (Figure 4A). Neither of the two stress-inducing factors alone induced significant alterations in RAD50 transcription. However, preincubation of cells with lower doses of HRE (10 and 25 μg/mL) in combination with oxidative stress resulted in a notable increase of 31% and 28% (*p* < 0.05), respectively, suggesting a synergistic effect of lower HRE doses and 10 mM H_2_O_2_. Irrespective of the type of stress applied, 50 μg/mL of HRE exhibited the most pronounced therapeutic effect on RAD50 expression, bringing it back to levels comparable to the control group (Figure 4B,C).

#### 2.5.3. Effects of HRE on BRCA1 Gene Expression and Radiosensitivity

At a concentration of 10 μg/mL, HRE significantly increased BRCA1 gene expression by approximately two-fold. However, higher concentrations of the extract showed no significant impact compared to control cells (Figure 4A). Treatment with H_2_O_2_ alone or combined with HRE did not alter BRCA1 gene expression shortly after treatment (Figure 4B). Of note, exposure to 2 Gy radiation resulted in a four-fold increase in BRCA1 mRNA levels, and preincubation with 10 μg/mL HRE further enhanced gene expression by 6.3 times compared to the control HRE^−^ γ-IR^−^ cells. Compared to the HRE-untreated, γ-irradiated group (HRE^−^ γ-IR^+^), pretreatment with HRE at 25 and 50 μg/mL concentrations significantly reduced BRCA1 transcription (Figure 4C). Importantly, pretreatment with ethanolic *Haberlea* extract above 10 μg/mL before 2 Gy γ-radiation significantly reduced BRCA1 transcription, bringing it to almost the control level.

#### 2.5.4. Influence of Ethanolic HRE on RAD51 Gene Expression

The RAD51 mRNA relative quantity remained unchanged at all tested concentrations of *Haberlea* ethanolic extract, showing no effect on gene expression compared to control cells (Figure 4A). Treatment with 10 mM H_2_O_2_ resulted in a significant 43% (1.75-fold) reduction in RAD51 transcript levels and preincubation with HRE did not alter gene expression levels (Figure 4B). In contrast, exposure to 2 Gy ionizing radiation led to a notable 45% up-regulation of RAD51 transcription. The addition of 10 μg/mL HRE before irradiation slightly further upregulated the gene by 1.72-fold compared to the HRE^−^ γ-IR^−^ calibrator sample. However, HRE at 25 μg/mL and 50 μg/mL concentrations restored RAD51 expression to near initial levels (Figure 4C).

#### 2.5.5. Modulation of CDKN1A Gene Expression by Ethanolic HRE and Cellular Stressors

Figure 4A illustrates that the transcription of the CDKN1A gene in cells treated with varying concentrations of ethanolic HRE alone closely resembles that of the double negative control group, HRE^−^ H_2_O_2_^−^/γ-IR^−^. Treatment with 10 mM H_2_O_2_ resulted in a significant 22% decrease in CDKN1A transcript content, further reduced when cells were pretreated with 10 μg/mL HRE (Figure 4B). Although the observed alterations were less than two-fold, statistical analysis confirmed their significance, indicating a cumulative effect of the combined treatment. Increasing the concentration of HRE used for cell pretreatment progressively restored CDKN1A gene expression toward that of non-treated control cells of the HRE^−^ H_2_O_2_^−^ group, with complete restoration observed at 50 μg/mL HRE (*p* > 0.05). Exposure to 2 Gy radiation led to a 47% increase in p21 mRNA levels, partially attenuated by preincubation with 50 μg/mL HRE (Figure 4C). While the observed differences in CDKN1A gene expression were not pronounced, an intriguing trend was observed. Both oxidative stress and γ-IR radiation exerted opposite effects on CDKN1A transcription. Within two hours after treatment, oxidative stress decreased CDKN1A transcription, while γ-IR radiation increased it. Notably, in both stress conditions, a trend towards normalisation of CDKN1A gene transcription was observed at 50 μg/mL HRE.

#### 2.5.6. Expression of Human BBC3 Gene Coding for p53 Upregulated Modulator of Apoptosis (PUMA)

The expression of the proapoptotic Bcl-2-binding component 3 (BBC3; also known as PUMA) encoding gene BBC3 was significantly downregulated when cells were incubated with the extract. The three tested HRE concentrations yielded a similar end effect—the depletion of the BBC3 transcript (Figure 4A). The addition of H_2_O_2_ to the culture medium resulted in a two-fold elevation of the BBC3 mRNA level (Figure 4B, red bar). When the cells were pretreated with HRE and then subjected to 10 mM H_2_O_2_, we observed a decrease in transcription from BBC3 at 10 and 25 µg/mL compared to the HRE^−^ H_2_O_2_^+^ cells, which, however, remained higher than the control level and an increase at 50 µg/mL HRE. As shown in Figure 4C, the cellular response to γ-ray exposure led to a 40% decrease in BBC3 transcription. Extract at 10 μg/mL did not change this picture, while administration of HRE at concentrations of 25 and 50 μg/mL resulted in a four- and six-fold increase in the expression of the gene coding for PUMA.

## 3. Discussion

Extracts of *H. rhodopensis* have been undoubtedly shown to contain a wide variety of biologically active substances, with myconoside being the most prevalent [6,17,18,19,23,38,39]. Consistent with these findings, myconoside was the predominant compound in the ethanol HRE used in the current study.

The effects of irradiation and oxidation on living systems have significant implications, causing damage and disruption to biological macromolecules and structures. This can decrease cell viability, leading to pathological conditions or even mortality [46]. DNA damage is particularly influential in these processes. It can manifest as single and double-strand breaks (SSBs and DSBs), base modifications, apurinic/apyrimidinic sites, cross-links, and the formation of adducts [47,48]. The detrimental impact on the DNA molecule caused by various chemical and physical factors is referred to as genotoxic effects. In radiotherapy, genotoxic effects on tumour cells are desirable, as they ideally induce apoptosis. However, it is essential to note that radiotherapy can also cause damage to healthy cells and even activate radioprotective mechanisms in specific cancerous cells. Previous studies have demonstrated that extracts from the endemic plant *Haberlea rhodopensis* exhibit radioprotective properties. However, the extent to which these extracts can be utilised in radiotherapy remains unclear. Therefore, investigating the precise molecular mechanisms through which these extracts confer radioprotection or radiosensitivity in cells is paramount.

### 3.1. HRE Protects HeLa Cells against Radiation and H_2_O_2_-Induced DNA Damage

Our experimental results indicated that preincubation of HeLa cells with an extract derived from *Haberlea rhodopensis* followed by exposure to 10 mM H_2_O_2_ or 2 Gy γ-radiation did not lead to significant changes in cellular morphology, granularity, or cell cycle progression (Appendix A). Thus, under the experimental conditions, no observable phenotypic alterations were induced in the studied cells during the initial hour post-treatment. However, it is essential to note that these treatments may still significantly impact the cellular molecular machinery. Our findings demonstrate that both 10 mM H_2_O_2_ and 2 Gy γ-IR caused DNA damage, as evidenced by the presence of single- or double-strand breaks and alkali-liable sites in the DNA, as detected by the Comet assay variant used in our experiments.

Notably, we observed that the preincubation of cells with HRE resulted in concentration-dependent alterations in the genotoxic effects of these treatments. Intriguingly, the effect of increasing concentrations of HRE on genotoxicity differed between the two stressors. The highest concentration of the extract exhibited the most pronounced mitigation of H_2_O_2_-induced damage, while the lowest concentration showed efficacy against radiation-induced DNA damage. These divergent responses may be attributed to the distinct mechanisms by which H_2_O_2_ and radiation exert their genotoxic effects. Nevertheless, our study revealed that 10 μg/mL and 50 μg/mL HRE concentrations protected against radiation and H_2_O_2_-induced DNA damage, respectively. These genoprotective effects of HRE may be attributed to its ability to scavenge free radicals generated during these treatments or facilitate the repair of DNA lesions, thereby preserving genome integrity [3,48].

### 3.2. Redox/Antioxidant Response to HRE and HRE Post-Applied Stress

An increase in SOD activity and LPO level manifested the early response of cells incubated with HRE alone. SOD (EC 1.15.1.1) is a pivotal cellular detoxification enzyme catalyzing the dismutation of the highly hazardous superoxide anion (*O_2_) into hydrogen peroxide (H_2_O_2_) and molecular oxygen (O_2_) [49]. It has been shown that at lower oxidative stress, SOD increased while high H_2_O_2_ (50 mM) inhibited it. Therefore, HRE itself acts as a weak oxidant, inducing SOD activities and LPO production, and the latter may result primarily from the interaction of HRE with cell membrane lipids. Recently, it has been shown that myconoside at a concentration of 5 μg/mL was not cytotoxic but could alter/disrupt the plasma membrane lipid order of the treated cells [50]. The glycoside myconoside is abundant in *Haberlea rhodopensis* plant and extracts, and the extracts used in the present study contained 1.4 μg, 3.5 μg, and 7 μg myconoside for 10, 25, and 50 μg/mL HRE, respectively. In addition, it has been demonstrated that flavonoids and polyphenols enriched in *Haberlea* extracts may exhibit dual activities and act as both antioxidants and prooxidants, leading to opposite outcomes [43,44,45]. This could explain the increased SOD activity and LPO content observed when cells were treated with HRE.

LPO products, reactive aldehydes, and lipid radicals formed during LPO can cause DNA damage, leading to genotoxic and mutagenic effects and eventually to the development of pathological conditions [51]. Therefore, the low genotoxicity detected in HRE-only-treated cells could result from the increased level of LPO. The increased SOD activity revealed a weak prooxidant activity of the applied HRE. That ultimately could induce an adaptive response to subsequent oxidative stress.

Further, our data revealed that regardless of the applied stress, H_2_O_2_ or radiation, the initial stress response in HeLa cells involved activation of SOD and CAT enzymes and increased LPO levels. In contrast, levels of tGSH and GPx activity decreased. Previous studies have demonstrated the preservation of antioxidant enzyme activity in desiccated *Haberlea* leaves, indicating the presence of functional enzymes in HRE [52], and the air-dried leaves retained significant activities of SOD and peroxidase [53]. Additionally, phenolic compounds in plant extracts can enhance the activity of antioxidant and phase II enzymes [6,54,55]. Our findings of elevated SOD activity and tGSH levels in cells pre-incubated with HRE, particularly at higher extract concentrations, align with these observations. Similar to those we found, decrements in GPx activity have been reported in tumour cells exposed to radiation or ellagic acid treatment alone or in combination. Ellagic acid, a natural phenol antioxidant found in fruits and vegetables, can induce ROS generation in HeLa cells, which further increases when combined with γ-radiation [56]. This corresponds to our observation of increased LPO levels in cells treated with HRE. HRE preincubation mitigated oxidative stress, most probably by scavenging the ROS, resulting in the detected reduced activity of SOD, CAT, and GPx antioxidant enzymes and LPO levels and an increase in tGSH content in comparison to the cells subjected to stress without HRE preincubation.

### 3.3. Gene Transcription Response to Stress after HRE Pre-Incubation

Critical genes, including ATM, DNA-PK, TP53, RAD50, and BRCA, are pivotal in the complex cellular DNA damage response network, regulating various factors and pathways involved in the cell cycle, apoptosis, DNA repair, metabolism, and senescence [46,57,58,59,60,61]. In this study, we investigated the expression of these genes in cells preincubated with HRE and exposed to radiation or H_2_O_2_ treatment. Our analysis focused on genes such as ATM, BRCA1, RAD50, RAD51, p21, and PUMA, which play critical roles in DNA repair signalling, homologous recombination repair, DSB DNA repair, HR, and apoptosis regulation [59,60,61,62,63]. The ATM kinase is activated by autophosphorylation at Ser-1981 and subsequently phosphorylates other DNA-damage response pathway components, including p53/TP53 and BRCA1 [60].

#### 3.3.1. Gene Activity upon HRE Administration Alone

ATM (Ataxia telangiectasia mutated), a serine/threonine protein kinase, is a DNA damage-responsive protein that activates the DNA damage response (DDR) checkpoint signalling [60]. The extract from *H. rhodopensis* did not affect, per se, the activity of the ATM and CDKN1A genes. At 10 μg/mL HRE, RAD50, and BRCA1 genes were upregulated (15% and two-fold, respectively), while at 25 and 50 μg/mL of the extract, the expression of these genes was restored to a near-control level, suggesting a weakly induced DDR at the low dose of the extract and a neutral effect at the higher doses. On the other hand, the transcript level of the BBC3 gene was reduced two-fold at all HRE concentrations applied, pointing out a significant down-regulatory effect on the proapoptotic regulator PUMA.

Recently, in a study on the molecular mechanism of the anti-cancerous potential of Morin (a mulberry leaf extract) in HeLa cells, it was reported that 48 h of treatment with the extract resulted in prominent downregulation of survivin genes. At the same time, the expression of p53 and p21 mRNAs increased between 20 and 80%, depending on the Morin concentration [64]. However, there was no data on Morin extract’s effects in its earlier application stages. In contrast, our previous and current data indicate no significant change in DNA damage response and repair genes TP53 [65], ATM, RAD51, and CDKN1A expression two hours after administration of HRE to HeLa cells, as well as concentration dependence of RAD50 and BRCA1 mRNA levels and apparent downregulation of the proapoptotic gene BBC3.

#### 3.3.2. Gene Activity upon Stressor Treatment

CDKN1A among the genes related to cell cycle checkpoints and BBC3 among the apoptosis-related genes were selected as valuable candidate biodosimeteric gene markers [66]. Accordingly, in our experiments, an increase in the relative mRNA concentrations of ATM, BRCA1, RAD51, and CDKN1A was detected upon exposure to 2 Gy radiation (50%, 4-fold, 45%, and 47%, respectively, compared to the control samples), which is an indication of the induction of DNA DSB signalling pathways. At the same time, BBC3 was downregulated by 40%, suggesting suppression of apoptosis soon after irradiation. Studies published by two research groups reported an opposite effect of γ-IR on the expression of the ATM gene. Irradiation of human blood with a 5 Gy radiation dose caused a 2-fold and 3.8-fold change in gene expression 30 min and 90 min post-irradiation, respectively [48]. At the same time, other authors showed a two-fold decrease in ATM mRNA level four hours after irradiation of mouse white blood cells with 2 Gy radiation [63]. The observed wide-range changes in ATM mRNA indicated that gene expression highly depends on the received radiation dose and the post-irradiation time points of analysis. A non-linear, more complex relationship between exposure doses, post-irradiation time points, cell type, and gene expression was reported for several other genes, e.g., RAD50, CDKN1A, and BBC3 [63,67]. In particular, the expression levels of BRCA and RAD50 genes two to five hours post-irradiation with a 2 Gy radiation dose have not changed significantly, while those of BBC3 and CDKN1A increased [63,67,68].

A different picture emerged when cells were treated with H_2_O_2_. The expression of ATM, RAD50, BRCA1, and BBC3 genes changed insignificantly compared to the control, while that of p21 was reduced by 22% (*p* < 0.05), and that of the RAD51 gene decreased by 43%. Our studies showed that in HeLa cells, two hours after irradiation with 2 Gy γ-IR, induction of the DNA repair signalling pathway has already begun by activating the transcription of the ATM, BRCA1, RAD51, and CDKN1A genes. At the same time, the BBC3 gene expression seems to be suppressed by γ-IR and activated early after the induction of oxidative stress. Considering all this, our results obtained by the chosen scheme of treatment of HeLa cells with radiation and H_2_O_2_ are not unexpected.

#### 3.3.3. Gene Activity upon Combined Preincubation with HRE and Treatment with Stressors

Exposure to IR, including radiation therapy used for cancer treatment, could induce a variety of DNA damage, including damage to the bases as well as SSBs and DSBs in the DNA backbone [69,70]. Of all the types of DNA damage, DSBs are the most dangerous to cell health and survival [71,72]. DSBs represent the most biologically significant lesions induced by IR, and the effectiveness of DNA DSB repair determines the cellular resilience to radiation [70,73]. Three general pathways contribute to the repair of IR-induced DSBs in mammalian cells: homologous recombination (HR), classic nonhomologous end-joining (cNHEJ), and the alternative NHEJ (aNHEJ) [69,74,75]. The protein product of the tumor suppressor BRCA1 gene acts as a key regulator of the three main DSB repair pathways, HR, cNHEJ, and aNHEJ, thereby maintaining genome integrity [70,76]. It has been demonstrated that BRCA1-mutant cancer cells have impaired DNA DSB repair and are particularly vulnerable to ionizing radiation, while the expression of BRCA1 restores the radioresistance [70,77,78]. Aside from BRCA1, defects in the DSB checkpoint and repair genes ATM and TP53 led to chromosomal instability and were associated with the tumour grade [78,79]. Accordingly, our analysis of the early cellular response to radiation stress revealed that cells exposed to 2 Gy γ-IR had higher mRNA levels of the DDR genes ATM, BRCA1, and TP53 [65]. This was most likely caused by a rise in DNA strand breaks, as detected via CA. Pre-administration of 10 μg/mL HRE led to further up-regulation of BRCA1 transcription (as well as TP53) and a significant attenuation and diminishing of DNA damage. Pre-incubation with 50 μg/mL HRE sets ATM, BRCA1, and TP53 gene expression to a near-control level; however, the DNA damage was found to remain as high as in the IR-exposed cells. These results pointed out the possible bidirectional action of the HRE extract, which is concentration-dependent. Further studies should determine the relevance and applicability of these findings in both radiation protection and radiation therapy.

The combination of lower HRE concentrations and a stressor produced cumulative effects on regulating ATM, RAD50, and CDKN1A transcription. In cells pre-incubated with 50 μg/mL HRE and then subjected to H_2_O_2_ stress, the relative mRNA levels of CDKN1A and BBC3 changed insignificantly, and those of ATM, RAD50, and CDKN1A genes were restored to a level comparable to that of the untreated control. At the same time, RAD51 remained downregulated at the level detected after the treatment with the stressor alone (about a 45% decrease compared to the untreated control). When cells were incubated with HRE before radiation exposure, we again detected differential effects of the low and higher extract concentrations. In cells pre-incubated with 10 μg/mL HRE, the expression of ATM, RAD51, and CDKN1A genes remained similar to that observed after 2 Gy irradiation, being upregulated by 60%, 72%, and 55% compared to the untreated control group. For the transcription of BRCA1, some cumulative effects of low HRE doses and radiation were detected.

Further, our data indicate that pretreatment with the higher amounts of HRE for one hour before genotoxic stress resulted in downregulation of gene expression (elevated in irradiated cells without extract supplementation), bringing the transcript levels of ATM, BRCA1, RAD50, and RAD51 to the control levels. Similar attenuation of the stress effect has been observed for the ATM gene after pretreatment with the three active compounds extracted from *Podophyllum hexandrum* (G-002M). The increased expression of ATM in human blood exposed to 5 Gy radiation was decreased by combined treatment with G-002M [48]. Hayrabedyan and co-authors reported a distinct pro-cell death and proapoptotic effect in cancer cells incubated with the *H. rhodopensis* extract for 24 h before oxidative or UV genotoxic stress [36]. Consistent with these data, a significant increase in the proapoptotic BBC3 gene transcript was detected due to incubation with higher amounts of HRE before irradiation. Most probably, it is an indication of HRE-triggered proapoptotic processes in the examined cancer HeLa cells. This effect could potentially increase the effectiveness of radiation therapy for cancer, but additional, more in-depth studies are needed to confirm it.

## 4. Materials and Methods

### 4.1. Chemicals and Reagents

General chemicals such as ethanol, NaCl, KCl, Na_2_HPO_4_, KH_2_PO_4_, NaOH, EDTA, acetic acid, N-laurylsarcosine, RNase A, and DMEM medium were purchased from Sigma-Aldrich, Co. (St. Louis, MO, USA). Solutions for FACS analyses, BD FACSFlow™, BD FACS™ Clean, and BD FACSRinse™ were obtained from Becton, Dickinson, and Company (Franklin Lakes, NJ, USA). All other chemicals used for specific analyses are described in the relevant subsection.

### 4.2. Haberlea rhodopensis Leaves Gathering and Plant Identification

After obtaining official permission from the Bulgarian Ministry of Environment and Waters, leaves were collected from *H. rhodopensis* plants growing in their natural habitat (a region close to the village of Bachkovo, Rhodope Mountains, Bulgaria, 41.9520 N, 24.8587 E). The collection was carried out by Assoc. Professors Borislav Popov, MD and Radoslav Radev, MD, under the supervision of a representative of the Regional Inspectorate for Environmental and Water Control, Plovdiv, Bulgaria. The botanical identification was completed at the Department of Pharmacognosy, Faculty of Pharmacy of the Medical University of Sofia, Bulgaria. Following the rule that the amount of active substances directly depends on the conditions under which medicinal plants grow, the samples were collected in the exact location only from late May to early June.

### 4.3. Haberlea rhodopensis Extract Preparation

The cut leaves were dried in the dark, at room temperature, for 1 month. The dry leaves were refined to 1 mm particles. The mixture was macerated in 70% ethyl alcohol for 48 h (Bulgarian Pharmacopoeia Roll 3, p. 218, d 20 = 0.887), followed by distillation of the ethanol in a vacuum evaporator to an extract/liquid phase ratio of 5:1. The obtained primary extract was further concentrated in a vacuum distillation apparatus of Ulbricht (residual pressure of 0.3 atmospheres and temperature up to 50 °C). The process was terminated when an azeotropic mixture of 5% ethanol and a volume ratio of 1:1 extract to extraction solvent was obtained. The crude extract was filtered through filter paper to remove emulsified chlorophyll and non-polar chemicals. The resultant extract was standardised according to the formula for determining the relative density, d20. The differences in the relative densities of the extract and the same volume of water at 20 °C were determined in g/cm^3^ using an analytical balance (with an accuracy of 10^−4^ g). Extracted substances ranged between 98 × 10^−3^ and 113 × 10^−3^ g/cm^3^ (average 105 × 10^−3^ g/cm^3^). Total stock extract with a 100 mg/mL concentration was diluted and used in the experiments.

### 4.4. High Performance Liquid Chromatography

The prepared total ethanol *Haberlea rhodopensis* leaf extract was analysed using the high performance liquid chromatographic (HPLC) method. The HPLC system used in our research consisted of a Waters 2487 dual λ absorbance detector, a Waters 1525 binary pump, and Breeze 3.30 software (Waters, Milford, MA, USA). The separation of molecules was achieved with a Kinetex^®^ column (100 mm × 2.1 mm, 5 μm; Phenomenex, Torrance, CA, USA) maintained at 26 °C with a flow rate of 0.6 mL/min. The obtained 100 mg/mL HRE (from Section 4.3) was diluted to 10 mg/mL with 70% ethanol and subjected to HPLC analysis according to the protocol previously utilised by Amirova et al. [39] with slight modifications. For myconaside determination, the eluents 2% acetic acid (Solvent A) and acetonitril (Solvent B) were used. The gradient’s final conditions are described in Appendix A. The injection volume of the samples was 20 µL. The research was conducted at the Department of Industrial Microbiology, Laboratory of Applied Biotechnologies, The Stephan Angeloff Institute of Microbiology, Bulgarian Academy of Science (Plovdiv, Bulgaria) in collaboration with Innova Ltd. (Sofia, Bulgaria).

### 4.5. Cells Culturing and Treatment

HeLa cells (Human cervix epithelial carcinoma, CCL-2™, ATCC^®^, Manassas, Virginia, United States) were cultured to confluence (1.6 × 10^6^ cells/mL) in six-well plates with 2 mL of DMEM medium supplemented with 10% foetal bovine serum at 37 °C and 5% CO_2_. Except for the controls, cell cultures were preincubated with 10, 25, or 50 μg/mL ethanol HRE for 1 h at 37 °C and then were exposed to 2 Gy γ-rays (Gammatron S-80 ^60^Co source at a dose rate of 89.18 cGy/min in a water bath, 37 °C; 1.25 MeV, HC-FMRP/USP; Siemens, Munich, Germany) or 10 mM H_2_O_2_ for 30 min at room temperature. In summary, four groups of cell samples were prepared for each stressor applied: double negative control (HRE^−^ γ-IR^−^/H_2_O_2_^−^); samples incubated with HRE alone (HRE^+^ γ-IR^−^/H_2_O_2_^−^); samples treated only with a genotoxic agent (HRE^−^ γ-IR^+^/H_2_O_2_^+^); and cells that have undergone both treatments with HRE and a stressor (HRE^+^ γ-IR^+^/H_2_O_2_^+^). Two hours following the irradiation or just after hydrogen peroxide treatment, cells were detached (by scraping) from the surface and harvested by centrifugation at 2500× *g* for 5 min, washed with 1 × PBS, pH 7.4, and subjected to flow cytometry, comet assay, redox component evaluation, and gene expression analyses.

### 4.6. Flow Cytometry

Rinsed cells were pelleted, fixed by adding ice-cold 75% ethanol, and stored at −20 °C overnight or until the analysis. Before flow cytometry, fixed cells were washed with 1 × PBS pH 7.4 and incubated with RNase A (to 0.1 mg/mL) for 30 min at 37 °C. After staining with 50 µg/mL of propidium iodide for 30 min at room temperature, 100,000 cells were analysed using the BD FACS Canto apparatus (Becton, Dickinson & Company, Franklin Lakes, NJ, USA) in the dark. The distribution of cells according to FSC-H, SSC-H, and DNA content (FL-2A) was performed to detect variations in the size, granularity, and cell cycle progression, respectively. The flow cytometry analysis software FlowJo™ (Becton, Dickinson & Company, Franklin Lakes, NJ, USA) was used to process the data.

### 4.7. Redox Components Assessment

The activities of antioxidant enzymes such as superoxide dismutase, catalase, and glutathione peroxidase, the concentration of the antioxidant glutathione, and malondialdehyde as an indicator of lipoperoxidation LPO, were determined in treated and control cells.

#### 4.7.1. Lipid Peroxidation Test

The lipoperoxidation was estimated by the amount of thiobarbituric acid (TBA) reactive substances (TBARS) according to the method of Hunter et al. [80]. A 1 mL aliquot of the cellular fraction (1 mg/mL protein) of each sample was incubated with 0.625 mL of 40% trichloroacetic acid/5 N HCl/2% TBA in 2:1:2 ratios at 100 °C for 10 min. After cooling and centrifugation at 2500× *g* for 10 min, the absorbance was read at 532 nm against the appropriate blank. The amount of TBARS was expressed in nanomoles of MDA per mg of protein (nmoL/mg protein), using a molar extinction coefficient of 1.56 × 10^5^ M^−1^cm^−1^.

#### 4.7.2. Total Glutathione Levels

tGSH levels were measured according to Tietze et al. [81]. The sulfhydryl groups of GSH present in the sample and those that resulted from the reduction of the oxidised glutathione (GSSG) in the presence of NADPH and glutathione reductase reacted with DTNB. The absorption peak of the yellow-coloured 5-thio-2-nitrobenzoic acid (TNB) was read at 412 nm. The values were calculated using GSSG as a reference standard and was expressed in ng/mg protein.

#### 4.7.3. Catalase Enzyme Activity

Catalase activity was determined according to the method of Aebi [82] via the decrease in absorption at 240 nm provoked by the enzymatic decomposition of H_2_O_2_. The activity of CAT was expressed as ΔA240/min/mg protein (U/mg protein) using a molar absorptivity of 43.6 M cm^–1^, and one unit is equal to the μmoles of hydrogen peroxide degraded per minute per mg of protein.

#### 4.7.4. Superoxide Dismutase Enzyme Assay

The activity of SOD was determined according to Beauchamp and Fridovich [83]. The superoxide radicals, generated photochemically, reduced the nitroblue tetrazolium (NBT) presented in the reaction mixture to insoluble blue formazan. The absorption of the colour product was measured at 560 nm, and values were expressed as U/mg protein. A unit of SOD activity is the amount of enzyme producing 50% inhibition of NBT reduction. Results are presented as SOD activation percentage.

#### 4.7.5. Glutathione Peroxidase Activity

The GPx activity assay performed here is an adaptation of the method of Paglia and Valentine [84]. Briefly, glutathione peroxidase activity was measured indirectly by a coupled reaction with glutathione reductase (GR). Oxidised glutathione GSSG, produced upon hydroperoxide reduction by GPx, is recycled to its reduced state by GR and NADPH. The oxidation of NADPH to NADP^+^ is accompanied by a decrease in absorbance at 340 nm. Under conditions where the GPx activity is rate-limiting, the decrease in the A_340_ is directly proportional to the GPx activity in the sample. The enzyme activity is expressed as nmoL oxidised NADPH/min/mg protein (U/mg protein).

### 4.8. Comet Assay (Single-Cell Gel Electrophoresis)

Genotoxicity of 10 mM H_2_O_2_ and 2 Gy γ-rays was determined by an alkaline comet assay. Agarose LE (Molecular Biology Grade) and TopVision Low Melting Point Agarose were purchased from Thermo Fisher Scientific Inc. (Waltham, MA, USA). After incubation with HRE, γ-rays, and H_2_O_2_, cells were collected by centrifugation at 2800× *g* for 5 min. Cells were then washed with 1 × PBS (137 mM NaCl, 2.7 mM KCl, 8.06 mM Na_2_HPO_4_, and 1.47 mM KH_2_PO_4_, pH 7.4), centrifuged as above, resuspended in 1 mL of the same buffer, and mixed with low-gelling agarose to a 0.7% (*w*/*v*) final concentration. The resulting agarose-cell suspension was spread on a pre-coated microscopic slide and sealed with a coverslip. The coverslip was removed after agarose solidification (4 °C for 10 min). The slides were maintained in a dark, cold room during all subsequent steps. The agarose-embedded cells were subjected to cell lysis (1 M NaCl; 50 mM EDTA, pH 8.0; 30 mM NaOH; 0.1% N-laurylsarcosine) for 60 min. To unwind dsDNA, the microgels were submerged in alkaline electrophoresis buffer (10 mM EDTA, pH 8.0; 30 mM NaOH) for 30 min. Then, the single-stranded DNA in the gel was subjected to electrophoresis at 0.45 V/cm for 20 min. The slides were rinsed with dH_2_O to neutralise the alkali in the gel, dehydrated by consecutive incubations in 75 and 95% ethanol for 5 min each, and left to air-dry.

Microgels containing treated and non-treated control cells were observed under an epi-fluorescent microscope Leitz, Orthoplan, VARIO ORTHOMAT 2 (Leica, Wetzlar, German) with a 450–490 nm bandpass filter using 250× magnification. Before fluorescence microscopy analysis, DNA was stained with SYBR Green I (Molecular Probes, Eugene, OR, USA). Comet assay data quantitation was carried out using TriTek Comet Score Freeware v1.5 software (TriTek, Corp., Sumerduck, VA, USA). Several parameters can be used to evaluate the alkaline CA results following the recommendations for statistical quantification of CA data described in [85]. The study used the Olive moment (OM) parameter for CA data representation. This parameter encompasses an integrative approach, incorporating both tail length and intensity during its calculation, thus offering comprehensive information regarding the genotoxicity of the substances under investigation [85].

### 4.9. Gene Transcription Assessment by RT-qPCR

#### 4.9.1. Total RNA Preparation and First-Strand cDNA Synthesis

Total RNA was extracted using the GeneJET™ RNA Purification Kit (Thermo Fisher Scientific Inc., Waltham, MA, USA) following the manufacturer’s protocol for mammalian cultured cells’ total RNA. The average A_260/280_ ratio was 2.06 ± 0.02, indicating the purification quality of the extracted RNAs. The obtained RNAs were treated with RNase-free Deoxyribonuclease I (DNase I, EURx Sp., Gdansk, Poland) at a concentration of 1 U/μg RNA in the presence of 1 U/µL RNase inhibitor (EURx Sp., Gdansk, Poland) at 37 °C for 30 min. The reaction was stopped by adding EDTA to a final concentration of 5 mM, and DNase I was heat-inactivated at 65 °C for 10 min.

Polyadenylated mRNAs were reverse transcribed using the oligo (dT)_20_ primer and the NG dART RT-PCR kit (EURx Sp., Gdansk, Poland) following the manufacturer’s instructions. The first strand cDNA was synthesized using 450 ng of total RNA as a template, and 1/30 of the reverse transcriptase reaction was used in the subsequent qPCR assays. The reverse transcriptase minus (RT) controls and no template controls (NTC) were also carried through.

#### 4.9.2. Real-Time PCR

The expression of several genes of interest, namely ATM, BRCA1, CDKN1A (p21), RAD50, RAD51, and BBC3 (PUMA), was evaluated by quantitative real-time PCR (qPCR). The housekeeping gene glyceraldehyde-3-phosphate dehydrogenase (GAPDH) was the endogenous control. The sequences of gene-specific primers (Eurofins Genomics, Ebersberg, Germany) used in this study are given in Table 1.

qPCR reactions were performed in a Rotor-Gene™ 6000 Real-time PCR thermal cycler (Corbett Life Science, Qiagen, Venlo, The Netherlands) using the SG qPCR Master Mix (2×) (EURx Sp., Gdansk, Poland) according to the manufacturer’s instructions. Following 10 min at 95 °C for hot-start polymerase activation, the two-step PCR was carried out for 45 cycles, including 15 s at 95 °C and 60 s at 60 °C. To examine the specificity of the polymerase reaction, melting curve (dF/dT) analysis was performed with ramping from 55 °C to 95 °C, rising by 1 °C each step, and waiting for 5 s before fluorescence acquisition. In addition, the correct size of the amplified products was analysed by agarose gel electrophoresis (1.8% agarose gel, 0.5 × TBE (44.5 mM Tris-borate, 1 mM EDTA; pH 8.3)).

For data quantification, the delta-delta Ct method (2^−ΔΔCt^) [86] and the two standard curves method were applied using the Rotor-Gene 6000 Series Software 1.7 (Corbett Life Science, Qiagen, Venlo, The Netherlands). Similar results were obtained regardless of which of the two methods for relative quantitative analysis was used. The expression of each gene was analysed in three independent experiments. The transcript levels in each sample were normalised to those of the reference gene GAPDH and calibrated to the respective double negative control sample HRE^−^ γ-IR^−^/H_2_O_2_^−^. In general, the expression of a particular gene is significantly changed (up- or down-regulated) if there is at least a two-fold difference in its mRNA quantity between the sample of interest and the calibrator (the quantity of the transcript in the latter one is considered as 1).

### 4.10. Statistical Analysis

SPSS Statistics (Statistical Package for Social Sciences; IBM, North Castle, NY, USA) version 19.0 software was used to analyse the significance of differences between the experimental groups. A student *t*-test (two-tailed significance) was carried out, performing a paired-samples *t*-test or independent-samples *t*-test depending on the compared samples. Values were reported as means ± SD of triplicate measurements; the significance level was 5%, and *p* < 0.05 was considered statistically significant.

## 5. Conclusions

The research presented in this study demonstrates, for the first time, the modulation of expression of several crucial stress-responsive genes when cells are treated with a proven radioprotective *H. rhodopensis* plant extract. Irrespective of the stressor applied, HeLa cells exhibited similar characteristics in their immediate cellular antioxidant response. Both hydrogen peroxide and gamma-irradiation stressors led to the generation of lipid peroxides and the activation of SOD and CAT enzymes while causing a decrease in the total glutathione level and glutathione peroxidase activity. In general, pretreatment of HeLa cells with HRE before stress exposure showed potential for partially alleviating the impact of the stressor on components of the cellular redox system, except for GPx activity. Regarding genotoxic stress, pretreatment with HRE demonstrated protective effects against genotoxicity induced by H_2_O_2_ and γ-IR. The genoprotective ability of HRE was found to be specific to the stressor applied, depending on the concentration of the extract. As the concentration of HRE increased, protection against H_2_O_2_-induced genotoxicity became more pronounced, while protection against genotoxicity caused by 2 Gy γ-radiation decreased. Regarding gene expression regulation, pretreatment with HRE at a concentration of 50 μg/mL exhibited the highest effectiveness in modulating the effects of stress induced by 10 mM H_2_O_2_ and 2 Gy irradiation. We demonstrated that the *Haberlea rhodopensis* plant extract, in which myconoside was identified as the most abundant compound, specifically influences the observed dynamics in gene expression. However, it is essential to acknowledge that the mechanisms underlying these changes are likely complex and would require further extensive experimentation. Consequently, future scientific papers will focus on a comprehensive understanding of the intricate biology governing the interaction between the bioactive extract and the cellular response. Further studies should determine the relevance and applicability of these findings in both radiation protection and radiation therapy.

## Figures and Tables

**Figure 1 ijms-24-15964-f001:**
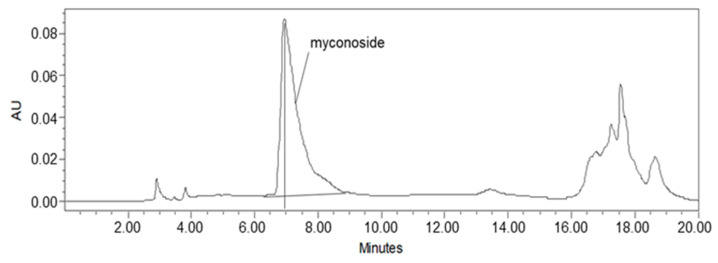
HPLC analysis of the ethanolic *Haberlea rhodopensis* leaves extract used in the study. The myconoside was identified as the most abundant compound in HRE, with a chromatographic retention time of 7 min.

**Figure 2 ijms-24-15964-f002:**
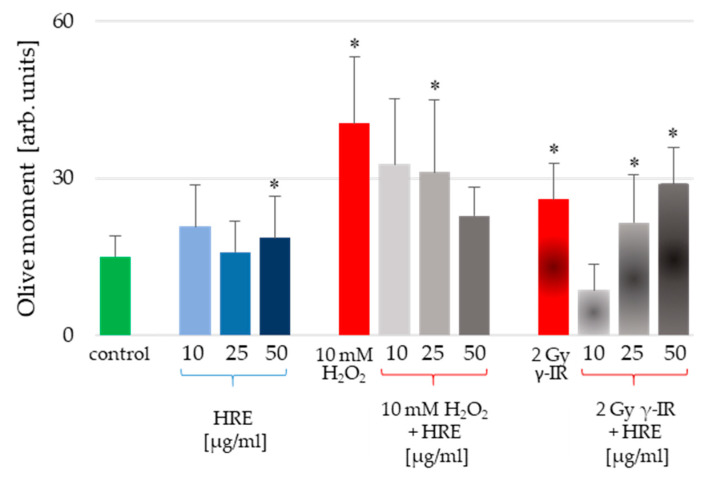
Genotoxicity assessment of HeLa cells incubated with HRE alone at a concentration of 10, 25, or 50 μg/mL (blue bars), cells pretreated with HRE and subjected to oxidative stress (10 mM H_2_O_2_) or ionizing radiation (2 Gy γ-IR) (grey bars); cells subjected to the stressor alone (red bars) and control cells without any treatment (green bar). Results are expressed as mean values of the Olive moment (OM) ± SD of triplicate measurements; the significance level was 5%, and * *p* < 0.05 vs. the control group.

**Figure 3 ijms-24-15964-f003:**
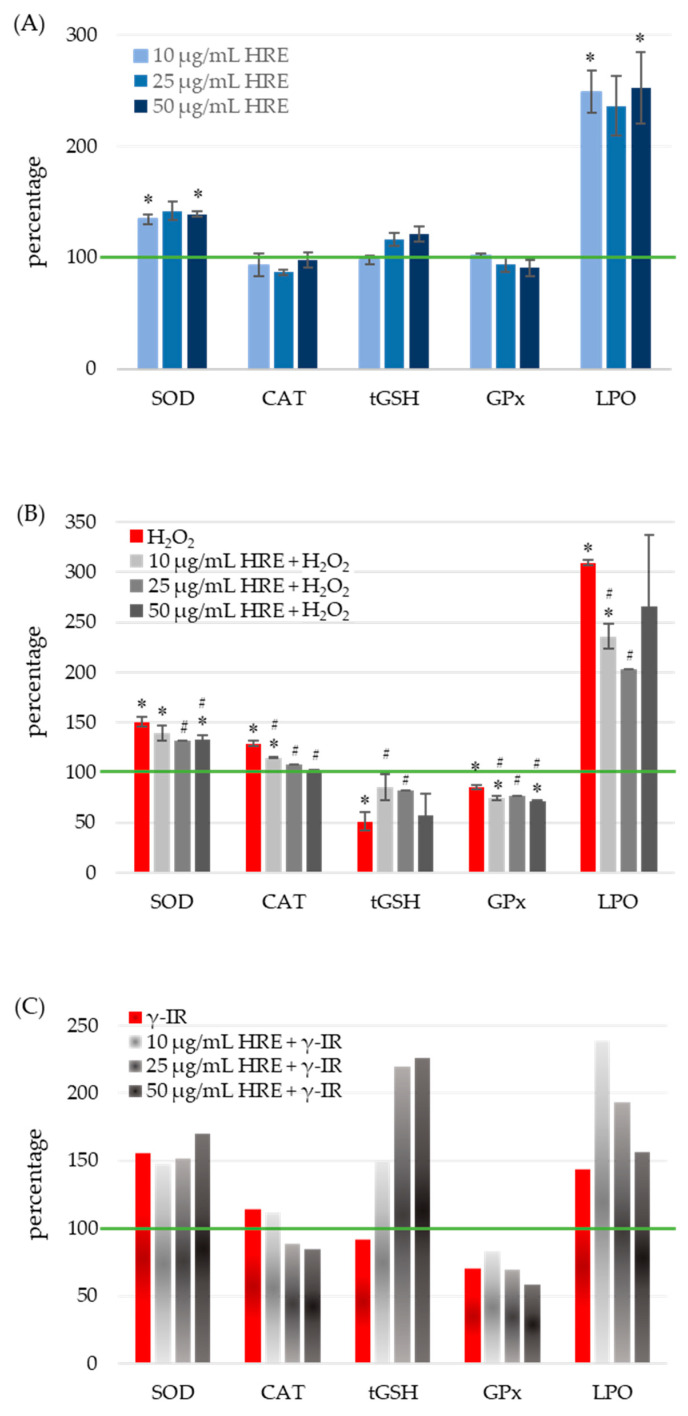
Response of cellular redox system components to treatment with HRE and a stressor within a short post-treatment period. HeLa cell cultures were handled as described in “Materials and Methods” Section 4.5 and assayed for SOD, CAT, GPx enzyme activities, MDA, and tGSH levels. The green line indicates the level or the activity of the respective redox component or enzyme in non-treated control cells. (**A**) Effect of incubation with HRE alone. *H. rhodopensis* extract was directly added to the culture media to a final concentration of 10, 25, or 50 μg/mL, and HeLa cells were incubated without any stressor. (**B**) Effect of H_2_O_2_ and the combination of HRE and H_2_O_2_ on cellular antioxidants and LPO. HeLa cell cultures were treated with 10 mM H_2_O_2_ to induce oxidative stress. (**C**) Effect of 2 Gy γ-IR with or without HRE preincubation on cellular antioxidants and LPO. Results are presented as a percentage of the corresponding control level, which is 100% (green line). * *p* < 0.05 vs. control group; # *p* < 0.05 vs. stressor (H_2_O_2_ or γ-IR) treated group.

**Figure 4 ijms-24-15964-f004:**
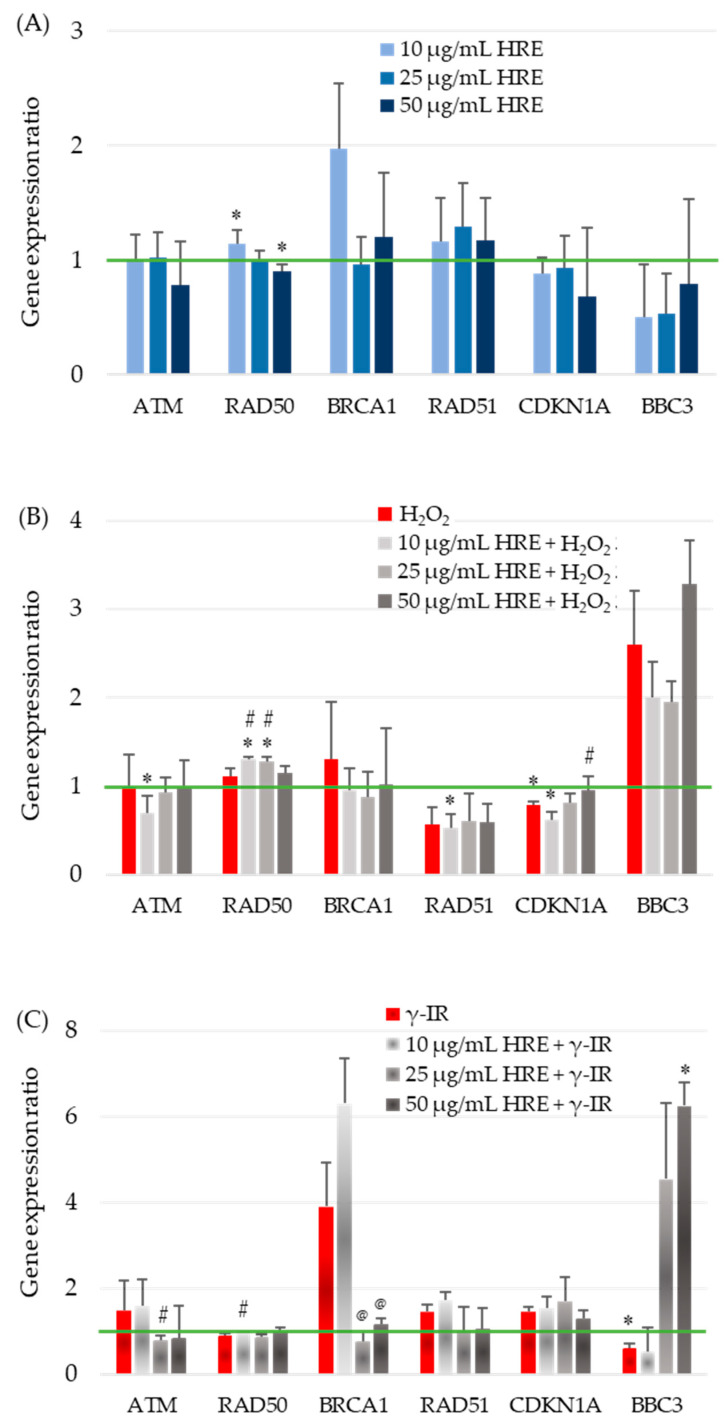
Evaluation of stress-responsive gene transcription in the first hours after HRE and stressor treatment. HeLa cell cultures were handled as described in “Materials and Methods” (Section 4.5), and the ATM, RAD50, BRCA1, RAD51, CDKN1A, and BBC3 relative transcript levels were assessed by RT-qPCR. (**A**) For gene transcription after incubation with HRE alone, 10, 25, or 50 μg/mL HRE extract was directly added to the culture media; HeLa cells were incubated without any stressor. (**B**) Relative gene expression in cells treated with 10 mM H_2_O_2_ and the combination of HRE and H_2_O_2_. (**C**) Gene transcription response to 2 Gy γ-IR, with or without HRE preincubation. Results are presented as the ratio of the corresponding control level, which is 1 (green line). * *p* < 0.05 vs. control group; # *p* < 0.05 vs. stressor (H_2_O_2_ or γ-IR) treated group; @ *p* < 0.05 vs. 10 + γ-IR treated group.

**Table 1 ijms-24-15964-t001:** Oligonucleotides used for priming of specific PCR gene amplification.

Gene	NCBI Ref Seq	Oligonucleotide Sequence 5′-3′	Amplicon, bp
ATM	NM_000051.3	For TGCTGTGAGAAAACCATGGAAGTGA Rev TCCGGCCTCTGCTGTAAATACAAAG	137
BRCA1	NM_007294.3	For CACCCAATTGTGGTTGTGCAGC Rev GTCCAGCTCCTGGCACTGGTAGAG	141
p21 (CDKN1A)	NM_000389.4	For AGAGGAAGACCATGTGGACCTGTCA Rev AGAAATCTGTCATGCTGGTCTGCC	134
BBC3 (PUMA)	NM_014417.4	For GGATGGCGGACGACCTCAA Rev GGGTAAGGGCAGGAGTCCCATG	119
RAD50	NM_005732.3	For TGGTGATGCTGAAGGGAGACACA Rev TTGTTGGCTCATCCAAGGCAATG	147
RAD51	NM_133487.3	For CAAGCATCAGCCATGATGGTAGAA Rev AGAAACCTGGCCAAGTGCATCTG	132
GAPDH	NM_002046.5	For ACCAGGTGGTCTCCTCTGACTTCAA Rev ACCCTGTTGCTGTAGCCAAATTCG	136

Abbreviations: ATM, Ataxia telangiectasia mutated; BRCA1, the gene for Breast cancer type 1 susceptibility protein; p21 also known as CDKN1A, cyclin-dependent kinase inhibitor 1; BBC3, Bcl-2-binding component 3, also known as PUMA; p53-upregulated modulator of apoptosis; RAD50 and RAD51, genes encoding DNA repair proteins 50 and 51; GAPDH, glyceraldehyde-3-phosphate dehydrogenase; bp, base pairs; For, forward; Rev, reverse.

## Data Availability

Not applicable.

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
