# Peer review of "Haberlea rhodopensis Extract Tunes the Cellular Response to Stress by Modulating DNA Damage, Redox Components, and Gene Expression"

_ijms, 2023, doi:10.3390/ijms242115964_

Round 1

Reviewer 1 Report (New Reviewer)

Comments and Suggestions for Authors

Staneva et al. have explored the impact of H. rhodopensis extract (HRE) on cellular responses to genotoxic and oxidative stress. Their manuscript stands out for its clear and professional use of the English language, which makes it easily comprehensible. Additionally, the authors have adeptly organized their results section, using well-structured figures to present their data. However, I have some additional suggestions for improving figure readability below. The authors have also created a comprehensive discussion section that delves into how HRE pretreatment affects gene activity. Furthermore, the manuscript includes an updated list of references. I've noted some minor points below:

  1. While it's possible to identify the myconoside peak in the HPLC chromatogram because it's the most abundant compound, I would recommend performing LC-MS analysis to definitively confirm this identification.
  2. Figure 3C lacks error bars.
  3. Consider revising the labels on the Y-axis of Figures 3A, 3B, and 3C. The current label, "% from control level," might not provide readers with the most informative context.
  4. It's advisable to provide a more in-depth discussion of the observation that "pretreatment with ethanolic Haberlea extract at concentrations exceeding 10 μg/mL before exposure to 2 Gy γ-radiation significantly reduces BRCA1 transcription".
  5. To emphasize the significance of their study, the authors should discuss potential real-world implications and relevance for further research. 
  6. Acknowledging any limitations of the current research and suggesting possible directions for future studies would be beneficial.

Author Response

We thank the reviewer for the valuable remarks and suggestions. Below are the point-by-point answers to the reviewer's comments:

While it's possible to identify the myconoside peak in the HPLC chromatogram because it's the most abundant compound, I would recommend performing LC-MS analysis to definitively confirm this identification.

Answer: The prepared total ethanol Haberlea rhodopensis leaves extract was analysed by High Performance Liquid Chromatographic (HPLC) method according to the protocol previously utilized by Amirova et al. [39]. For myconaside determination using HPLC, the eluents 2% acetic acid (Solvent A) and acetonitrile (Solvent B) were used. The gradient final conditions are described in Table S1. The resulting peak in the chromatogram is quite definitive and clear. Further analyses, e.g. LC-MS, were not carried out since no other substances were detected around this peak.

  1. Amirova, K.M.; Dimitrova, P.A.; Marchev, A.S.; Krustanova, S.V.; Simova, S.D.; Alipieva, K.I.; Georgiev, M.I. Biotechnologically-Produced Myconoside and Calceolarioside E Induce Nrf2 Expression in Neutrophils. Int J Mol Sci 2021, 22, doi:10.3390/Ijms22041759

Figure 3C lacks error bars.

Answer: Figure 3C represents the effect of 2 Gy γ-IR with or without HRE preincubation on cellular antioxidants and LPO. We acknowledge that Figure 3C represents results from a single experiment and, unfortunately, could not repeat those experiments due to experimental complications and lack of funds. However, it is essential to note that the significance of these results was cautiously discussed, considering their limited scope. Still, we believe that its presence here is vital to follow the logic of the other results and provide further research directions. We believe that the findings presented in this particular graphic (just one out of the 13 included) do not alter the overall conclusions drawn from our study.

Consider revising the labels on the Y-axis of Figures 3A, 3B, and 3C. The current label, "% from control level," might not provide readers with the most informative context.

 Answer: Concerning the reviewer's recommendation, we have edited the labels on the Y-axis of Figures 3A, 3B, and 3C to "percentage". In the figure legend, we specified that the results are presented as a percentage of the level of the corresponding control sample.

It's advisable to provide a more in-depth discussion of the observation that "pre-treatment with ethanolic Haberlea extract at concentrations exceeding 10 μg/mL before exposure to 2 Gy γ-radiation significantly reduces BRCA1 transcription".

Answer: We thank the reviewer for this suggestion. The paragraph discussing this observation was included in the manuscript's main text.

Exposure to IR, including radiation therapy used for cancer treatment, could induce a variety of DNA damage, e.g. damage to the bases as well as SSBs or DSBs in the DNA backbone (Mahaney et al., 2007; Kan & Zhang, 2015). Of all the types of DNA damage, DSBs are the most dangerous to cell health and survival (Khanna KK, Jackson, 2001; Pfeiffer et al., 2004). Therefore, DSBs represent the most biologically significant lesions induced by IR, and the effectiveness of DNA DSBs repair determines the cellular resilience to radiation (Mladenov et al., 2013; Kan & Zhang, 2015). Three general pathways contribute to the repair of IR-induced DSBs in mammalian cells: homologous recombination (HR), the classic non-homologous end-joining (cNHEJ) and the alternative NHEJ (aNHEJ) (Mahaney et al., 2007; Jeggo et al., 2011; Boboila et al., 2012; Xie et al., 2009). The protein product of the tumour suppressor BRCA1 gene acts as a key regulator of the three main DSB repair pathways, HR, cNHEJ and aNHEJ, thereby maintaining genome integrity (Deng, 2006; Kan & Zhang, 2015). It has been demonstrated that BRCA1-mutant cancer cells have impaired DNA DSB repair and are particularly vulnerable to ionising radiation, while the expression of BRCA1 restores the radioresistance (Abbott et al., 1999; Ding et al., 2004; Kan & Zhang, 2015). Aside from BRCA1, defects in the DSB checkpoint and repair genes ATM and TP53 led to chromosomal instability and were associated with the tumour grade (Ding et al., 2004; Cao et al., 2006). Accordingly, our analysis of the early cellular response to radiation stress revealed that cells exposed to 2 Gy γ-IR had higher mRNA levels of the DDR genes ATM, BRCA1, and TP53 (Dimitrova et al., 2023). This was most likely caused by increased DNA strand breaks, as detected via CA. Pre-administration of 10 μg/mL HRE led to further up-regulation of BRCA1 (as well as TP53) transcription and a significant attenuation and diminishing of DNA damage. Preincubation with 50 μg/mL HRE sets ATM, BRCA1 and TP53 gene expression to near control level, however, the DNA damage was found to remain as high as in the IR-exposed cells. These results pointed out the possible bidirectional action of the HRE extract, which is concentration-dependent. Further studies should determine the relevance and applicability of these findings in both radiation protection and radiation therapy.

Abbott, Derek W., Marilyn E. Thompson, Cheryl Robinson-Benion, Gail Tomlinson, Roy A. Jensen, Jeffrey T. Holt. 1999. BRCA1 Expression Restores Radiation Resistance in BRCA1-defective Cancer Cells through Enhancement of Transcription-coupled DNA Repair*, Journal of Biological Chemistry, 1999, 274(26): 18808-18812.

Boboila C., F.W. Alt, B. Schwer. Classical and alternative end-joining pathways for repair of lymphocyte-specific and general DNA double-strand breaks. Adv Immunol, 116 (2012), pp. 1-49

Cao Liu, Sangsoo Kim, Cuiying Xiao, Rui-Hong Wang, Xavier Coumoul, Xiaoyan Wang, Wen Mei Li, Xiao Ling Xu, Joseph A De Soto, Hiroyuki Takai, Sabine Mai, Stephen J Elledge, Noboru Motoyama, Chu-Xia Deng. 2006. ATM–Chk2–p53 activation prevents tumorigenesis at an expense of organ homeostasis upon Brca1 deficiency. The EMBO Journal (2006) 25:2167-2177; https://doi.org/10.1038/sj.emboj.7601115

Deng CX (2006) BRCA1: cell cycle checkpoint, genetic instability, DNA damage response and cancer evolution. Nucleic Acids Res 34: 1416–1426

Dimitrova, N., Staneva D., Popov, B., Alexandrova, A., Georgieva, M., Miloshev, G. Haberlea rhodopensis alcohol extract normalizes stress-responsive transcription of the human TP53 gene. Journal of Experimental Biology and Agricultural Sciences 2023, 11, 405-415, doi:http://dx.doi.org/10.18006/2023.11(2).405.415.

Ding, S., Sheu, L., Yu, J. et al. Abnormality of the DNA double-strand-break checkpoint/repair genes, ATM, BRCA1 and TP53, in breast cancer is related to tumour grade. Br J Cancer 90, 1995–2001 (2004). https://doi.org/10.1038/sj.bjc.6601804

Jeggo PA, Geuting V, Lobrich M. The role of homologous recombination in radiation-induced double-strand break repair. Radiother Oncol 2011;101:7-12.

Kan, C. & Zhang, J. 2015. BRCA1 Mutation: A Predictive Marker for Radiation Therapy? Int J Radiation Oncol Biol Phys, Vol. 93, No. 2, pp. 281e293, 2015

Khanna KK, Jackson SP. DNA double-strand breaks: Signaling, repair and the cancer connection. Nat Genet 2001;27:247-254.

Mahaney BL, Meek K, Lees-Miller SP. Repair of ionizing radiation induced DNA double-strand breaks by non-homologous end-joining. Biochem J 2009; 417:639-650.

Mladenov E, Magin S, Soni A, et al. DNA double-strand break repair as determinant of cellular radiosensitivity to killing and target in radiation therapy. Front Oncol 2013;3:113.

Pfeiffer P, Goedecke W, Kuhfittig-Kulle S, et al. Pathways of DNA double-strand break repair and their impact on the prevention and formation of chromosomal aberrations. Cytogenet Genome Res 2004; 104:7-13.

Xie A., A. Kwok, R. Scully. Role of mammalian Mre11 in classical and alternative nonhomologous end joining. Nat Struct Mol Biol, 16 (2009), pp. 814-818

To emphasize the significance of their study, the authors should discuss potential real-world implications and relevance for further research.

Answer: We thank the reviewer for this comment.

Radioprotective effects of Haberlea rhodopensis extracts (HRE), both in vitro and in vivo, were previously extensively reported. However, most of them focused on and reported the final result, i.e., the endpoint effect of HRE treatment, without revealing the molecular mechanisms underlying the observed effects on living cells. The present research aimed to shed some light on the molecular and cellular mechanisms that lead to the observed effects of H. rhodopensis extracts. The results indicate that HRE extract modulates HeLa cellular response to stress by regulating the efficiency of DNA damage response, adjusting the level/activity of redox components, and changes in gene expression. These findings will help better understand the reasons behind the observed final point (real world) effect of HRE treatment. The findings need further investigation, and those are our current efforts to provide a detailed map of all players in these HRE modulating functions in HeLa cells in normal and extreme conditions like under radiotherapy.

Acknowledging any limitations of the current research and suggesting possible directions for future studies would be beneficial.

Answer: The research presented in this study demonstrates, for the first time, the integrated effect of a proven radioprotective Haberlea rhodopensis plant extract on several studied stress response molecular mechanisms: DNA damage, cellular redox homeostasis, and gene transcription. The modulation of expression from several crucial stress-responsive genes in genotoxin-exposed cells after HRE pre-treatment was detected. However, it is essential to acknowledge that the mechanisms underlying these changes are complex and would require further extensive experimentation. Consequently, future scientific papers will focus on a comprehensive understanding of the intricate biology governing the interaction between the bioactive extract and cellular response. In this paper, we aim to provide a comprehensive overview of the observed dynamics and trends in stress-responsive gene expression under this treatment with a focus on the early cellular response to pre-treatment with HRE, two hours after exposure to a genotoxin. For definitive elucidation of the mechanisms involved, examining the long-term alterations in redox, gene expression and DNA damage statuses is also relevant. In addition, the consequences of the two genotoxins would be repaired following particular cellular pathways. To depict a comprehensive picture of HRE molecular mechanisms of action upon radioactive and oxidative stress, other players of the stress response (sensors and effectors) must also be examined.

Reviewer 2 Report (New Reviewer)

Comments and Suggestions for Authors
  1. Dear authors, I have several remarks about:
  2. I. " Materials and Methods":

  3. 1. Chemicals and reagents should be included and described.
  4. 2. Why you did not include a description of the HPLC method, which you have included in the Results?
  5. You must include this method in the Materials and Methods section and you must provide the chromatographic conditions. Something more you should provide a reference for this method (if it was already validated) or if you have developed this method you should provide data about the method validation.
  6. II. Results
  7. Figure 1. The pick of the analysed compound (myconoside) is almost 2 minutes wide. This is not normal. You should provide a better chromatogram.
    1.  

Author Response

We thank the reviewer for the valuable comments. We have addressed them below.

" Materials and Methods": Chemicals and reagents should be included and described.

General chemicals and reagents are described in a new subsection, 4.1. Chemicals and reagents.

Please see the text below, also included in the main text of the manuscript.

4.1. Chemicals and Reagents

General chemicals such as ethanol, NaCl, KCl, Na2HPO4, KH2PO4, NaOH, EDTA, acetic acid, N-laurylsarcosine, RNase A and DMEM medium were purchased from Sigma-Aldrich, Co. (St. Louis, MO, USA). Solutions for FACS analyses, BD FACSFlow™, BD FACS™ Clean and BD FACSRinse™ were obtained from Becton, Dickinson and Company (Franklin Lakes, New Jersey, USA). All other chemicals used for specific analyses are described in the relevant subsection.

 More specific chemicals used in a given assay are described in the relevant subsection, e.g.,

 4.9.1. Total RNA Preparation and First-strand cDNA Synthesis

Total RNA was extracted using the GeneJET™ RNA Purification Kit (Thermo Fisher Scientific Inc, Waltham, Massachusetts, USA) following the manufac-turer's protocol for mammalian cultured cells' total RNA.

The obtained RNAs were treated with RNase-free Deoxyribonuclease I (DNase I, EURx Sp., Gdansk, Poland) at a concentration of 1 U/μg RNA in the presence of 1 U/µL RNase inhibitor at 37°C for 30 min.

Polyadenylated mRNAs were reverse transcribed using oligo (dT)20 primer and NG dART RT-PCR kit (EURx Sp., Gdansk, Poland)…

etc

Why you did not include a description of the HPLC method, which you have included in the Results? You must include this method in the Materials and Methods section and you must provide the chromatographic conditions. Something more you should provide a reference for this method (if it was already validated) or if you have developed this method you should provide data about the method validation.

In the previous version of the manuscript, the application of the HPLC method was described in the section Materials and Methods, subsection 4.2 Haberlea rhodopensis extract preparation, and chemical characterization.

In the revised manuscript, the description of the HPLC is provided in a separate subsection, 4.4. High Performance Liquid Chromatography, together with the reference and experimental conditions used.

Please see the text below, also included in the main body of the manuscript.

4.4.  High Performance Liquid Chromatography

The prepared total ethanol Haberlea rhodopensis leaves extract was analysed by High Performance Liquid Chromatographic (HPLC) method. The HPLC system used in our research consisted of a Waters 2487 dual λ absorbance detector, a Waters 1525 binary pump, and Breeze 3.30 software (Waters, Milford, USA). The separation of molecules was achieved with a Kinetex® column (100 mm × 2.1 mm, 5 μm; Phenomenex, Torrance, CA, USA) maintained at 26°C with a flow rate of 0.6 mL/min. The obtained 100 mg/mL HRE (from subsection 4.2.) was diluted to 10 mg/mL with 70% ethanol and subjected to HPLC analysis according to the protocol previously utilized by Amirova et al. [39] with slight modifications. For myconaside determination, the eluents 2% acetic acid (Solvent A) and acetonitrile (Solvent B) were used. The gradient final conditions are described in Table S1. The injection volume of the samples was 20 µL. The research was conducted at the Department of Industrial Microbiology, Laboratory of Applied Biotechnologies, The Stephan Angeloff Institute of Microbiology, Bulgarian Academy of Science, (Plovdiv, Bulgaria) in collaboration with Inova Ltd (Sofia, Bulgaria).

II. Results: Figure 1. The pick of the analysed compound (myconoside) is almost 2 minutes wide. This is not normal. You should provide a better chromatogram.

The prepared total ethanol Haberlea rhodopensis leaves extract was analysed by High Performance Liquid Chromatographic (HPLC) method according to the protocol previously utilized by Amirova et al. [39]. For myconaside determination using HPLC, the eluents 2% acetic acid (Solvent A) and acetonitril (Solvent B) were used. The gradient final conditions are described in Table S1. The resulting peak in the chromatogram is of this size because of the dilution used. Further dilution of the sample would have resulted in a narrower peak, but this was not carried out as no other substances were detected around this peak.

Amirova, K.M.; Dimitrova, P.A.; Marchev, A.S.; Krustanova, S.V.; Simova, S.D.; Alipieva, K.I.; Georgiev, M.I. Biotechnologically-Produced Myconoside and Calceolarioside E Induce Nrf2 Expression in Neutrophils. Int J Mol Sci 2021, 22, doi:10.3390/Ijms22041759.

This manuscript is a resubmission of an earlier submission. The following is a list of the peer review reports and author responses from that submission.

Round 1

Reviewer 1 Report

Comments and Suggestions for Authors

This study aims to investigate radioprotective and antioxidant properties of Haberlea rhodopensis  extract. The early response of HeLa cancer cells to gamma ionizing radiation (γ-IR) and oxidative stress after preincubation of the cells with  Haberlea rhodopensis extract was investigated. 

As HeLa cells is a cancer cell line, using these as a model for radioprotective  properties might be not appropriate to test the hypothesis. In cancer treatment, radiosensitization effect of natural products is preferable to enhance the efficiency of the treatment. As shown in many previous report, many types of cancer cells generate defense system including DNA repairing and anti-oxidant to survive after radiation. It therefore likely to be curious that whether the extract may decrease the efficiency of radiotherapy rather than radioprotective in cancer patients. 

Previous studies already reported radioprotective effects of this plant both in vitro and in vivo, this study therefore dose not really present novel concept. Moreover, the results are not consistent, the discussion is not clear and poor organized.

The first paragraph of page 4 should be removed. It dose not integrative describe the rationale or concept of this work. Just explain the functions and related pathways of each biomolecule.

Figure 2C, Are there any statistic difference and also SD?

Only comet assay was performed to reveal the anti-genotoxicity ability of the plant extract, it likely to be preliminary data which is not sufficient for the publication as research article. 

Discussion Section should be rewritten to integrative summarize and combine the finding from overall study. 

Characterization of the extract must be performed for the standardization of the extraction and the source of plant sample. Moreover, the author must determine the level of candidate bioactive compound(s) or at least perform qaunlitative measurement.

Author Response

Open Review 1

Quality of English Language

( ) I am not qualified to assess the quality of English in this paper
( ) English very difficult to understand/incomprehensible
( ) Extensive editing of English language required
( ) Moderate editing of English language required
( ) Minor editing of English language required
(x) English language fine. No issues detected

Yes

Can be improved

Must be improved

Not applicable

Does the introduction provide sufficient background and include all relevant references?

( )

(x)

( )

( )

Are all the cited references relevant to the research?

( )

(x)

( )

( )

Is the research design appropriate?

( )

( )

( )

(x)

Are the methods adequately described?

( )

(x)

( )

( )

Are the results clearly presented?

( )

( )

(x)

( )

Are the conclusions supported by the results?

( )

( )

(x)

( )

Comments and Suggestions for Authors

This study aims to investigate radioprotective and antioxidant properties of Haberlea rhodopensis extract. The early response of HeLa cancer cells to gamma ionizing radiation (γ-IR) and oxidative stress after preincubation of the cells with Haberlea rhodopensis extract was investigated. 

  1. As HeLa cells is a cancer cell line, using these as a model for radioprotective properties might be not appropriate to test the hypothesis. In cancer treatment, radiosensitization effect of natural products is preferable to enhance the efficiency of the treatment. As shown in many previous report, many types of cancer cells generate defense system including DNA repairing and antioxidant to survive after radiation. It therefore likely to be curious that whether the extract may decrease the efficiency of radiotherapy rather than radioprotective in cancer patients.

We thank the reviewer for this remark. Indeed, it has relevance, and we shall consider it in our future investigations of the extracts. To date, we would like to cordially explain why we have chosen the HeLa cell line in our studies.

HeLa cells are one of the most famous cervical cancer cell lines, used in many radiobiological studies. Importantly, in HeLa cells, the gene TP53 is not mutated (HeLa p53+/+) (Hoppeseyler & Butz, 1993; Leroy et al., 2014), and therefore, we have checked its expression levels in response to radiotherapy and Haberlea extracts supplementation.

Concerning the reviewer’s comment, we have edited the title of the manuscript. Its current title is: “Haberlea rhodopensis extract modulates the cellular response to stress by tuning DNA damage, redox components and gene expression”. The word and meaning of radioprotection are omitted in the text. We also stress the biological role of the plant extracts to our main findings: modulating cellular response to stress by tuning DNA damage, redox components and gene expression.

Tumours, incl. HeLa cell lines display heterogeneity, consisting of radioresistant and radiosensitive cells. Increased radioresistance has been frequently observed in tumours and cultured cells following repeated exposure to ionizing radiation (Muta & Koiwai, 1970, Koiwai & Muta, 1974). Traditionally, 1.8-2 Gy (Gray) of radiation delivered daily for five days per week is regarded as the conventional treatment scheme for carcinoma cervix in humans (Das et al., 2015). This repeated irradiation could induce radioresistance or an adaptive response to the first dose (Joiner, 1994). However, this is not the case in our study, as the cells were exposed only once to 2 Gy of gamma radiation; therefore, no induced resistance is expected. Furthermore, we used H2O2 as a second genotoxic agent, and the two stressors exposure, H2O2 and γ-IR, resulted in a statistically significant elevation of DNA damage.

It is worth noting that other authors have proven that the non-malignant HEK293 cell line and the prostatic cancer cell line PC3 responded similarly to treatment with H. rhodopensis methanol extracts mimicking ROS-induced apoptotic effect (Hayrabedyan et al., 2014).

  1. Previous studies already reported radioprotective effects of this plant both in vitro and in vivo, this study therefore dose not really present novel concept.

Yes, the radioprotective effects of Haberlea rhodopensis extracts (HRE), both in vitro and in vivo, were previously reported. However, most of them focused on and reported the final result, i.e., the endpoint effect of HRE treatment, without revealing the molecular mechanisms underlying the observed effects on living cells. The present research aimed to shed some light on the molecular and cellular mechanisms that lead to the observed effects of H. rhodopensis extracts. The results indicate that HRE extract modulated HeLa cellular response to stress by regulating the efficiency of genes involved in DNA damage response modulated the redox components, and led to changed gene expression.

These results need further investigation, and those are our current efforts to provide a detailed map of all players in these HRE modulating functions in HeLa cells in normal and extreme conditions like under radiotherapy.

  1. Moreover, the results are not consistent, the Discussion is not clear and poor organized.

 We agree that the results are not entirely consistent, but this is the case with many other scientific works, especially those observing the biological effects of plant extracts on complex models like tumour cell lines. We, therefore, did our best to explain our results consistently and pave the road to an increased interest in these extracts and future well-designed experiments based on the current results and knowledge.

The Discussion was rewritten to allow better representation and Discussion of the results.

  1. The first paragraph of page 4 should be removed. It dose not integrative describe the rationale or concept of this work. Just explain the functions and related pathways of each biomolecule.

The paragraph is deleted. Thank you for this suggestion, it indeed increased text clarity and reduced its heaviness for the reader.

  1. Figure 2C, Are there any statistic difference and also SD?

Effect of 2 Gy γ-IR with or without HRE preincubation on cellular antioxidants and LPO. We could not repeat those results due to experimental complications and lack of funds. Still, we believe that its presence here is important to follow the logic of the other results and provide further research directions.

  1. Only comet assay was performed to reveal the anti-genotoxicity ability of the plant extract, it likely to be preliminary data which is not sufficient for the publication as research article.

The Comet assay (Single-cell gel electrophoresis, SCGE), is a susceptible tool for assessment of damage in the DNA molecule at single cell level/base. The advantages of the Comet assay include its ability to detect genotoxicity in any cell type from both in vivo and in vitro sources and to identify DNA-damaging compounds at low concentrations. As such, Comet assay has been widely used for testing the genotoxicity of physical and chemical agents, materials, and biological compounds, including herbal substances and preparations as well as pharmaceuticals for human use (Arora et al., 2005; Verschaeve, 2015; Bajpayee et al., 2019; Grozdanova et al., 2020; Cardoso et al., 2022). In addition, the FDA Guidance on Industrial Genotoxicity Testing S2(R1) and Data Interpretation for Pharmaceutical Products Intended for Human Use (FDA, 2012) referred to SCGE as a method of choice. According to the many articles published using Comet assay, it is obvious that it is a basic tool in genotoxicological studies.

Therefore, the SCGE is fully recognized as a relevant assay for assessing DNA damage and evaluating induced genotoxicity and antigenotoxic property of various treatments. Due to the limited manuscript volume and size, we could not provide repetitions of experiments with other methods. We hope the reviewer understands this and will grant us with a positive evaluation.

  1. Discussion Section should be rewritten to integrative summarize and combine the finding from overall study.

 It has been rewritten and edited. Thank you for this suggestion.

  1. Characterization of the extract must be performed for the standardization of the extraction and the source of plant sample. Moreover, the author must determine the level of candidate bioactive compound(s) or at least perform qualitative

Following the rule that the amount of active substances directly depends on the conditions under which medicinal plants grow, the samples were collected in the same location only from late May to early June. The cut leaves were dried in the dark, at room temperature, for 1 month. The dry leaves were refined to 1 mm particles. The mixture was macerated in 70% ethyl alcohol for 48 hours (Bulgarian Pharmacopoeia Roll 3, p.218, d 20 = 0.887), followed by distillation of the ethanol in a vacuum evaporator to a drug/liquid phase ratio of 5:1. The obtained primary extract was further concentrated in a vacuum-distillation apparatus of Ulbricht (residual pressure of 0.3 atmospheres and temperature up to 50°C). The process was terminated when obtaining an azeotropic mixture of 5% ethanol and a volume ratio of 1:1 extract to drugs. The crude extract was filtered through filter paper to remove emulsified chlorophyll and non-polar chemicals. The resultant extract was standardized according to the formula for determining the relative density d20. The differences in the relative densities of the extract and the same volume of water at 20°C using an analytical balance to the nearest 10-4 g were determined in g/cm3. Extracted substances ranged between 9.8×10-3 and 11.3×10-3 g/cm3 (average 10.5×10-3 g/cm3). Total stock extract with a 100 mg/mL concentration was diluted and used in the experiments.

Studies on Haberlea rhodopensis have identified myconoside as one of its main active ingredients. The presence of myconoside in the used total ethanol Haberlea rhodopensis extract was quantified by HPLC to a content of 140 ± 2.3 mg/g dry weight. The analyses were performed on the HPLC system Waters 2487 using a dual λ absorbance detector, Waters 1525 binary pump (Waters, Milford, USA), and Breeze 3.30 software. The research was conducted at the Department of Industrial Microbiology, Laboratory of Applied Biotechnologies, The Stephan Angeloff Institute of Microbiology, Bulgarian Academy of Science, Plovdiv, Bulgaria. The separation of the molecules was carried out with a column Kinetex® (100 × 2.1 mm, 5 μm; Phenomenex; Torrance, CA, USA flow 0.6 mL/min), maintained at 26°C. The myconoside determination was based on an HPLC protocol previously used by Amirova et al., 2021.

The phenylethanoid glycoside myconoside [β-(3,4-dihydroxyphenyl)-ethyl-3,6-di-O-β-D-apifuranosyl-4-O-α,β-dihydrocaffeoyl-O-β-D-glucopyranoside] was identified as one of the main biologically active molecules in H. rhodopensis. Amirova and co-authors reported that the pure myconoside (32 µg/mL) and the fraction B (200 µg/mL) containing predominantly myconoside both increased the intracellular level of the Nrf2 protein (the nuclear factor erythroid 2 p45-related factors 2 (Nrf2), a transcriptional regulator of the cellular redox balance) in BM neutrophils and improved neutrophil survival (Amirova et al., 2021).

References

Amirova KM, Dimitrova PA, Marchev AS, Krustanova SV, Simova SD, Alipieva KI, Georgiev MI. Biotechnologically-Produced Myconoside and Calceolarioside E Induce Nrf2 Expression in Neutrophils. Int J Mol Sci. 2021 Feb 10;22(4):1759. doi: 10.3390/ijms22041759. PMID: 33578811; PMCID: PMC7916618.

Arora, S., Brits, E., Kaur, S., Kaur, K., Sohi, R. S., Kumar, S., & Verschaeve, L. (2005). Evaluation of genotoxicity of medicinal plant extracts by the comet and VITOTOX tests. Journal of environmental pathology, toxicology and oncology: official organ of the International Society for Environmental Toxicology and Cancer, 24(3), 193–200. https://doi.org/10.1615/jenvpathtoxoncol.v24.i3.50

Bajpayee, M., Kumar, A., & Dhawan, A. (2019). The Comet Assay: Assessment of In Vitro and In Vivo DNA Damage. Methods in molecular biology (Clifton, N.J.)2031, 237–257. https://doi.org/10.1007/978-1-4939-9646-9_12

Cardoso R, Dusinska M, Collins A, Manjanatha M, Pfuhler S, Registre M. and Elespuru R (2022) In vivo Mammalian Alkaline Comet Assay: Method Adapted for Genotoxicity Assessment of Nanomaterials. Front. Toxicol 4:903896. doi: 10.3389/ftox.2022.903896

Das S, Singh R, George D, Vijaykumar TS, John S. Radiobiological Response of Cervical Cancer Cell Line in Low Dose Region: Evidence of Low Dose Hypersensitivity (HRS) and Induced Radioresistance (IRR). J Clin Diagn Res. 2015 Jun;9(6):XC05-XC08. doi: 10.7860/JCDR/2015/14120.6074. Epub 2015 Jun 1. PMID: 26266200; PMCID: PMC4525590.

FDA (2012) Guidance for Industry S2(R1) Genotoxicity Testing and Data Interpretation for Pharmaceuticals Intended for Human Use. http://www.fda.gov/downloads/Drugs/GuidanceComplianceRegulatoryInformation/Guidances/UCM074931.pdf.

Grozdanova, T., Trusheva, B., Alipieva, K., Popova, M., Dimitrova, L., Najdenski, H., Zaharieva, M. M., Ilieva, Y., Vasileva, B., Miloshev, G., Georgieva, M., & Bankova, V. (2020). Extracts of medicinal plants with natural deep eutectic solvents: enhanced antimicrobial activity and low genotoxicity. BMC chemistry, 14(1), 73. https://doi.org/10.1186/s13065-020-00726-x

Hayrabedyan S., Todorova K., Zasheva D., Moyankova D., Georgieva D., Todorova J., Djilianov D. Haberlea rhodopensis has potential as a new drug source based on its broad biological modalities. Biotechnol. Biotechnol. Equip, 2014;27:3553–3560

Hoppeseyler, F., & Butz, K. (1993). Repression of Endogenous P53 Transactivation Function in Hela Cervical-Carcinoma Cells by Human Papillomavirus Type-16 E6, Human Mdm-2, and Mutant P53. Journal of Virology, 67(6), 3111-3117. https://doi.org/Doi 10.1128/Jvi.67.6.3111-3117.1993

Joiner, M. C. (1994). Induced Radioresistance: An Overview and Historical Perspective. International Journal of Radiation Biology, 65(1), 79–84. doi:10.1080/09553009414550111

Koiwai, S., & Muta, N. (1974). Studies on Radioresistance with HeLa Cells: Radiosensitivity and Chromosome Constitution. Radiation Research, 59(3), 717–723. https://doi.org/10.2307/3574088

Leroy, B., Girard, L., Hollestelle, A., Minna, J. D., Gazdar, A. F., & Soussi, T. (2014). Analysis of TP53 Mutation Status in Human Cancer Cell Lines: A Reassessment. Human Mutation, 35(6), 756-765. https://doi.org/10.1002/humu.22556

Muta, N., & Koiwai, S. (1970). Studies on Radioresistance with HeLa Cells. Establishment of a Radioresistant Line. Radiation Research, 43(2), 332–340. https://doi.org/10.2307/3573038

Verschaeve, Luc. 2015. Genotoxicity and Antigenotoxicity Studies of Traditional Medicinal Plants: How Informative and Accurate are the Results? Natural Product Communications, 10(8):1489-1493

Reviewer 2 Report

Comments and Suggestions for Authors

This manuscript has scientific merit, but needs to include more experiments. These issues should be addressed:

1- Time of collection of plant leaves and Voucher specimen needs to be issued.

2- Line 539: drug/liquid phase proportion of 5:1. Which drug? what does this mean? clarify?

3- The manuscript lacks any characterization of the leaves extract. MDPI and other journals require that publications on a plant extract perform structural characterization of plant extracts. The authors should show HPLC, GC_MS, LC-MS, or NMR analysis of thie extract. Having already published on this plant does not mean that this specific plant extract should be chemically characterized.

3- The authors need to test the protective effect of the extract in another cell line. At least another cancer cell line in addition to 1 "normal" human cell line.

4- Line 411-412 “Our data (Figure 2) revealed that regardless of the applied stress, SOD, CAT, and  LPO levels increased, while levels of tGSH and GPx decreased.  You should check your results. Only SOD is significantly increased. Discuss why other anti-oxidant enzymes did not show significance?

5- ATM and BRCA1 should be checked at protein levels. Phosphorylation status of ATM can be checked as well.

Author Response

Open Review 2

Quality of English Language

(x) I am not qualified to assess the quality of English in this paper
( ) English very difficult to understand/incomprehensible
( ) Extensive editing of English language required
( ) Moderate editing of English language required
( ) Minor editing of English language required
( ) English language fine. No issues detected

Yes

Can be improved

Must be improved

Not applicable

Does the introduction provide sufficient background and include all relevant references?

(x)

( )

( )

( )

Are all the cited references relevant to the research?

(x)

( )

( )

( )

Is the research design appropriate?

( )

(x)

( )

( )

Are the methods adequately described?

( )

(x)

( )

( )

Are the results clearly presented?

(x)

( )

( )

( )

Are the conclusions supported by the results?

(x)

( )

( )

( )

Comments and Suggestions for Authors

The comments are in the pdf attached: peer-review-30171654.v1.pdf

  1. Comet assay: If you have tail intensity values, please include them (also see Comment â„–8).

Comment â„–8: Line 630-1: "Several parameters were used to evaluate the alkaline CA results following the recommendations for statistical quantification of CA data described in [99]" - if several parameters were used, what not including tail intensity? it would be much more valuable to have that data here:

According to the CometScore Tutorial (http://AutoComet.com), several algorithms include the comet tail intensity parameter, e.g., Olive Moment (OM), total tail intensity, mean tail intensity, tail intensity integral, tail moment, etc. The Olive Moment and tail intensity are measures used in the Comet assay to assess DNA damage. Here's a comparison of the value and utility of each measure, along with their pros and cons:

The Olive Moment is a measure that combines both the length and intensity of DNA migration in the Comet assay. It allows

Comprehensive measure: It considers the extent of DNA migration (tail length) and the intensity of DNA damage (tail intensity).

Reflects different aspects of DNA damage: The Olive Moment provides a more holistic view of DNA damage, capturing both single- and double-strand breaks.

Sensitive to changes: It is often considered more sensitive than tail intensity alone, allowing for detecting subtle differences in DNA damage.

Tail intensity measures the intensity of DNA migration (the comet tail) in the Comet assay. Its advantages are the following:

Simple calculation: Tail intensity is relatively straightforward, as it involves measuring the intensity of fluorescence or staining in the tail region of the comet.

Easy interpretation: Tail intensity directly measures DNA damage intensity, making it easier to interpret and compare between samples.

Faster analysis: Analysing tail intensity values is generally quicker and more straightforward than calculating Olive Moment values.

But Tail intensity has drawbacks in the face of:

Limited information: Tail intensity alone does not capture the length or extent of DNA migration, potentially missing out on valuable information about DNA damage.

Insensitive to changes in tail length: Since tail intensity does not consider the length of the tail, it may fail to detect differences in DNA damage when tail length remains constant, but the intensity varies.

This comparison between the two commonly used parameters in CA data analytics will allow the reviewer to understand our choice in CA data evaluation.

  1. Comet assay Fig. 1+ Line 111: Explain why there are significant differences in only HRE treatment from the control values in green

Considering the apparent increase of OM provoked by the stressors, the 17% rise detected for 50 μg/mL HRE could be considered a common genotoxic effect. Similarly, using Comet assay, low genotoxicity was also detected for other plant extracts (Arora et al., 2005; Grozdanova et al., 2020). Citation: "This is not considered contradictory because DNA damage in the alkaline comet assay may not be permanent and hence may not necessarily result in mutations" (Arora et al., 2005). Cellular DNA repair systems could repair the low level of DNA damage and would not be dangerous. On the contrary, this low genotoxicity could prepare the cell to respond adequately to the subsequent genotoxic stress.

The early response of cells incubated with HRE was manifested by increased SOD activity and LPO level. SOD (EC 1.15.1.1) is a pivotal cellular detoxification enzyme catalysing the dismutation of the highly hazardous superoxide anion (*O2) into hydrogen peroxide (H2O2) and molecular oxygen (O2) (Fridovich, 1995). It has been shown that at lower oxidative stress, SOD increases while high H2O2 (50 mM) inhibits SOD (an H2O2-producing enzyme). On the other hand, it has been demonstrated that flavonoids and polyphenols, enriched in Haberlea extracts, may exhibit dual activities and act as both antioxidants and prooxidants, issuing opposite outcomes (Fujisawa et al., 2004; Metodiewa et al., 1999; Sergediene et al., 1999). Therefore, the HRE is a weak oxidant inducing SOD activities and LPO production.

LPO products, reactive aldehydes and lipid radicals formed during LPO can cause DNA damage leading to genotoxic and mutagenic effects and eventually to the development of pathological conditions (Gentile et al., 2017). Accordingly, the detected low genotoxicity in HRE-only treated cells could result from an increased level of LPO that, together with the increased SOD activity, revealed a weak prooxidant activity of the applied HRE. That ultimately could induce an adaptive response to subsequent oxidative stress.

  1. Figure 1. Genotoxicity Comet assay: why is the star not here? According to the statistical analysis, the difference between the control and the treated groups (10 μg/mL HRE and 50 μg/mL HRE + 10 mM H2O2) is insignificant.

                                         SD instead of STDV – it was corrected.

  1. Line 111: not true…" Notably, all tested concentrations of HRE did not induce genotoxicity in the cells (Figure 1, blue bars)".

We thank the reviewer for stressing this discrepancy in the description of the results. We carefully rewrote the paragraph to reflect the results obtained accurately:

Lines 111-119: Notably, incubation with HRE at 10 and 25 μg/mL did not induce cell genotoxicity. Only the highest tested concentration of 50 μg/mL HRE caused a statistically significant increase in the Olive moment critical parameter (Figure 1, blue bars). Conversely, the two genotoxic stressors (H2O2 and γ-IR) caused DNA damage. The OM values were significantly higher in genotoxin-treated cells than untreated control cells, exhibiting an average increase of 2.66-fold for H2O2-treated cells and 1.73-fold for γ-IR-treated cells. Considering the apparent increase of OM provoked by the stressors, the 17% raise detected for 50 μg/mL HRE could be considered a common genotoxic effect.

  1. Figure 2. Redox system components:
    • Units are missing (for 10, 25, 50) – units for the concentration of HRE, 10 μg/mL HRE, 25 μg/mL HRE and 50 μg/mL HRE, were added in the three panels of Fig. 2.
    • 2 Gy instead of γ-IR – With all due respect to the reviewer's remark, γ-IR was not changed to 2 Gy. Only the type of stress is given in the figure; the dose is specified in the Figure legend. The same is for H2O2; the stressor concentration is indicated in the Figure legend but not in the figure itself.

      Figure 3. Stress-responsive gene transcription:

  • How can a ratio have arbitrary units?

Thank you for the remark. The Y-axis title was changed to "Gene expression ratio."

 Units are missing (for 10, 25, 50) - units for the concentration of HRE, 10 μg/mL HRE, 25 μg/mL HRE and 50 μg/mL HRE, were added in the three panels of Fig. 3.

  1. Line 578: "centrifugation at 2500 rpm ": turn it into g, or otherwise write equipment, manufacturer, city, state

 2500 g, edited in the text as well. Thank you for this suggestion too.

  1. Line 630-1: "Several parameters were used to evaluate the alkaline CA results following the recommendations for statistical quantification of CA data described in [99]" - if several parameters were used, what not including tail intensity? It would be much more valuable to have that data here

 By this sentence, we cite the possibility of different parameters to be used to quantify the Comet assay results. We have choosen to present our results by the Olive Moment and proved the logic behind this by answering Question No 1.

References

Arora, S., Brits, E., Kaur, S., Kaur, K., Sohi, R. S., Kumar, S., & Verschaeve, L. (2005). Evaluation of genotoxicity of medicinal plant extracts by the comet and VITOTOX tests. Journal of environmental pathology, toxicology and oncology: official organ of the International Society for Environmental Toxicology and Cancer24(3), 193–200. https://doi.org/10.1615/jenvpathtoxoncol.v24.i3.50

Fridovich, I. Superoxide Radical and Superoxide Dismutases. Annu Rev Biochem 1995, 64, 97-112, doi:DOI 10.1146/annurev.bi.64.070195.000525

Fujisawa, S.; Atsumi, T.; Ishihara, M.; Kadoma, Y. Cytotoxicity, ROS-generation activity and radical-scavenging activity of curcumin and related compounds. Anticancer Res 2004, 24, 563-569.

Gentile F, Arcaro A, Pizzimenti S, Daga M, Cetrangolo GP, Dianzani C, Lepore A, Graf M, Ames PRJ, Barrera G. DNA damage by lipid peroxidation products: implications in cancer, inflammation and autoimmunity. AIMS Genet. 2017 Apr 18;4(2):103-137. doi: 10.3934/genet.2017.2.103. PMID: 31435505; PMCID: PMC6690246.

Grozdanova, T., Trusheva, B., Alipieva, K., Popova, M., Dimitrova, L., Najdenski, H., Zaharieva, M. M., Ilieva, Y., Vasileva, B., Miloshev, G., Georgieva, M., & Bankova, V. (2020). Extracts of medicinal plants with natural deep eutectic solvents: enhanced antimicrobial activity and low genotoxicity. BMC chemistry, 14(1), 73. https://doi.org/10.1186/s13065-020-00726-x

Metodiewa, D.; Jaiswal, AK; Cenas, N.; Dickancaite, E.; Segura-Aguilar, J. Quercetin may act as a cytotoxic prooxidant after its metabolic activation to semiquinone and quinoidal product. Free Radical Bio Med 1999, 26, 107-116, doi:Doi 10.1016/S0891-5849(98)00167-1.

Sergediene, E.; Jonsson, K.; Szymusiak, H.; Tyrakowska, B.; Rietjens, I.M.; Cenas, N. Prooxidant toxicity of polyphenolic antioxidants to HL-60 cells: description of quantitative structure-activity relationships. FEBS letters 1999, 462, 392-396, doi:10.1016/s0014-5793(99)01561-6.

Reviewer 3 Report

Comments and Suggestions for Authors

my comments are in the pdf attached,  if you have tail intesnity values, please include them, and explain why there are significant differences in only HRE treatment from the control values in green

Author Response

Open Review 3

Quality of English Language

( ) I am not qualified to assess the quality of English in this paper
( ) English very difficult to understand/incomprehensible
( ) Extensive editing of English language required
( ) Moderate editing of English language required
( ) Minor editing of English language required
(x) English language fine. No issues detected

Yes

Can be improved

Must be improved

Not applicable

Does the introduction provide sufficient background and include all relevant references?

( )

(x)

( )

( )

Are all the cited references relevant to the research?

( )

(x)

( )

( )

Is the research design appropriate?

( )

(x)

( )

( )

Are the methods adequately described?

( )

(x)

( )

( )

Are the results clearly presented?

( )

(x)

( )

( )

Are the conclusions supported by the results?

( )

( )

(x)

( )

Comments and Suggestions for Authors

This manuscript has scientific merit but needs to include more experiments. These issues should be addressed:

1- The time of collection of plant leaves and Voucher specimen needs to be issued.

We thank the reviewer for this remark. Following this recommendation we have provided all necessary details. After obtaining official permission from the Bulgarian Ministry of Environment and Waters, leaves were collected from H. rhodopensis plants growing in their natural habitat (a region close to the village of Bachkovo, Rhodope Mountains, Bulgaria, 41.9520 N, 24.8587 E). The collection was carried out by Assoc. Professors Borislav Popov, MD and Radoslav Radev, MD, under the supervision of a representative of the Regional Inspectorate for Environmental and Water Control, Plovdiv, Bulgaria. The certificate is included. The botanical identification was done at the Department of Pharmacognosy, Faculty of Pharmacy of the Medical University of Sofia, Bulgaria.

Following the rule that the amount of active substances directly depends on the conditions under which medicinal plants grow, the samples were collected in the same location only from late May to early June.

2- Line 539: drug/liquid phase proportion of 5:1. Which drug? what does this mean? clarify?

The collected fresh leaves were dried in the dark at room temperature for 1 month. The dry leaves were then refined to 1 mm particles and macerated in 70% ethyl alcohol for 48 hours (Bulgarian Pharmacopoeia Roll 3, p.218, d 20 = 0.887), followed by distillation of the ethanol in a vacuum evaporator to a drug material/liquid phase ratio of 5:1. The obtained primary extract was further concentrated in a vacuum-distillation apparatus of Ulbricht (residual pressure of 0.3 atmospheres and temperature up to 50°C). The process was terminated when obtaining an azeotropic mixture of 5% ethanol and a volume ratio of 1:1 extract to drug material. The crude extract was filtered through filter paper to remove emulsified chlorophyll and non-polar chemicals. The resultant extract was standardized according to the formula for determining the relative density d20. The differences in the relative densities of the extract and the same volume of water at 20°C using an analytical balance to the nearest 10-4 g were determined in g/cm3. Extracted substances ranged between 9.8×10-3 and 11.3×10-3 g/cm3 (average 10.5×10-3 g/cm3). Total stock extract with a 100 mg/mL concentration was diluted and used in the experiments.

3- The manuscript lacks any characterization of the leaf extract. MDPI and other journals require publications on a plant extract to perform structural characterization of plant extracts. The authors should show HPLC, GC_MS, LC-MS, or NMR analysis of this extract. Having already published this plant does not mean this specific plant extract should be chemically characterized.

Studies on Haberlea rhodopensis have identified myconoside as one of its main active ingredients. The presence of myconoside in total Haberlea rhodopensis extract was quantified by HPLC to 140 ± 2.3 mg/g dry weight. The analyses were performed on the HPLC system Waters 2487 using a dual λ absorbance detector, Waters 1525 binary pump (Waters, Milford, USA), and Breeze 3.30 software. The research was conducted at the Department of Industrial Microbiology, Laboratory of Applied Biotechnologies, Stephan Angeloff Institute of Microbiology, Bulgarian Academy of Science, Plovdiv, Bulgaria. The separation of the molecules was carried out with a column Kinetex® (100 × 2.1 mm, 5 μm; Phenomenex; Torrance, CA, USA flow 0.6 mL/min), maintained at 26°C. The myconoside determination was based on an HPLC protocol previously used by Amirova KM et al., 2021.

All these details are added in the text.

4- The authors need to test the protective effect of the extract in another cell line. At least another cancer cell line and 1 "normal" human cell line.

Here, we have provided our results with the famous human cell line HeLa, which has turned out to be a reference cell line over the years. Studying the molecular mechanisms (genes) by which different cell lines (cancerous or normal) would react to the treatment can be the subject and continuation for another experimental design and article.

5- Line 411-412 "Our data (Figure 2) revealed that regardless of the applied stress, SOD, CAT, and  LPO levels increased, while levels of tGSH and GPx decreased.  You should check your results. Only SOD is significantly increased. Discuss why other antioxidant enzymes did not show significance.

We carefully checked the results, and indeed, after treatment of cells with a stressor alone, the activity of SOD and CAT and the levels of LPO increased. In contrast, the tGSH and GPx activity level decreased compared to the non-treated control cells (green line). All these changes were statistically significant.

An increase in SOD activity and LPO level manifested the early response of cells incubated with HRE. SOD (EC 1.15.1.1) is a pivotal cellular detoxification enzyme catalyzing the dismutation of the highly hazardous superoxide anion (*O2) into hydrogen peroxide (H2O2) and molecular oxygen (O2) (Fridovich, 1995). It has been shown that at lower oxidative stress, SOD increases while high H2O2 (50 mM) inhibits SOD (an H2O2-producing enzyme). Therefore, the HRE is a weak oxidant inducing SOD activities and LPO production. The latter may result primarily from the interaction of HRE with cell membrane lipids. Recently, it has been shown that myconoside at a concentration of 5 μg/ml was not cytotoxic but could alter/disrupt the plasma membrane lipid order of the treated cells (Kostadinova et al., 2022). The glycoside myconoside is abundant in Haberlea rhodopensis plant and extracts, and the extracts used in the present study contained 1.4 μg, 3.5 μg and 7 μg myconoside for 10, 25 and 50 μg/mL HRE, respectively. In addition, it has been demonstrated that flavonoids and polyphenols, enriched in Haberlea extracts, may exhibit dual activities and act as both antioxidants and prooxidants, issuing opposite outcomes (Fujisawa et al., 2004; Metodiewa et al., 1999; Sergediene et al., 1999). This could explain the increased SOD activity and LPO content observed when cells were treated with HRE.

LPO products, reactive aldehydes and lipid radicals formed during LPO can cause DNA damage leading to genotoxic and mutagenic effects and eventually to the development of pathological conditions (Gentile et al., 2017). Therefore, the detected low genotoxicity in HRE-only treated cells could result from the increased level of LPO that, together with the increased SOD activity, revealed a weak prooxidant activity of the applied HRE. That ultimately could induce an adaptive response to subsequent oxidative stress.

6- ATM and BRCA1 should be checked at protein levels. The phosphorylation status of ATM can be checked as well.

We generally agree with the reviewer as this is a very relevant suggestion. However, in the present study, we focused on the cellular response at the transcriptional level to study whether the early stress response includes activation of genes involved in DDR, cell cycle control and apoptosis.

The respective protein levels or/and their enzyme activity can be shown in a consecutive study.

References

Amirova KM, Dimitrova PA, Marchev AS, Krustanova SV, Simova SD, Alipieva KI, Georgiev MI. Biotechnologically-Produced Myconoside and Calceolarioside E Induce Nrf2 Expression in Neutrophils. Int J Mol Sci. 2021 Feb 10;22(4):1759. doi: 10.3390/ijms22041759. PMID: 33578811; PMCID: PMC7916618.

Fridovich, I. Superoxide Radical and Superoxide Dismutases. Annu Rev Biochem 1995, 64, 97-112, doi:DOI 10.1146/annurev.bi.64.070195.000525

Fujisawa, S.; Atsumi, T.; Ishihara, M.; Kadoma, Y. Cytotoxicity, ROS-generation activity and radical-scavenging activity of curcumin and related compounds. Anticancer Res 2004, 24, 563-569.

Gentile F, Arcaro A, Pizzimenti S, Daga M, Cetrangolo GP, Dianzani C, Lepore A, Graf M, Ames PRJ, Barrera G. DNA damage by lipid peroxidation products: implications in cancer, inflammation and autoimmunity. AIMS Genet. 2017 Apr 18;4(2):103-137. doi: 10.3934/genet.2017.2.103. PMID: 31435505; PMCID: PMC6690246.

Kostadinova, A., Hazarosova, R., Topouzova-Hristova, T. et al. Myconoside interacts with the plasma membranes and the actin cytoskeleton and provokes cytotoxicity in human lung adenocarcinoma A549 cells. J Bioenerg Biomembr 54, 31–43 (2022). https://doi.org/10.1007/s10863-021-09928-x

Metodiewa, D.; Jaiswal, A.K.; Cenas, N.; Dickancaite, E.; Segura-Aguilar, J. Quercetin may act as a cytotoxic prooxidant after its metabolic activation to semiquinone and quinoidal product. Free Radical Bio Med 1999, 26, 107-116, doi:Doi 10.1016/S0891-5849(98)00167-1.

Sergediene, E.; Jonsson, K.; Szymusiak, H.; Tyrakowska, B.; Rietjens, I.M.; Cenas, N. Prooxidant toxicity of polyphenolic antioxidants to HL-60 cells: description of quantitative structure-activity relationships. FEBS letters 1999, 462, 392-396, doi:10.1016/s0014-5793(99)01561-6.

Round 2

Reviewer 1 Report

Comments and Suggestions for Authors

As mentioned in last review, about the model used in this study is not appropriate, Figure2C is from only an experiment without any repeat which can not confirm this will be reproducible. Moreover, authors' response can not address the novelty issue. To be published as research article, the exact underlying mechanisms must be determined.

Author Response

Open Review 1, Revision 2

 We are grateful to the reviewer for all provided comments. All are addressed carefully in our reply.

 Remark: As mentioned in last review, about the model used in this study is not appropriate, Figure2C is from only an experiment without any repeat which cannot confirm this will be reproducible.

Answer: We acknowledge that Figure 2C represents results from a single experiment, as we previously confirmed in our response to the reviewer. However, it is important to note that the significance of these results was cautiously discussed, considering their limited scope. It is our belief that the findings presented in this particular graphic (which is just one out of the 13 included) do not alter the overall conclusions drawn from our study.

Regarding the concern about the appropriateness of the model used (HeLa cell line), we have already addressed this issue comprehensively in our previous reply to the reviewers.

Remark: Moreover, authors' response cannot address the novelty issue. To be published as research article, the exact underlying mechanisms must be determined.

Answer: The research presented in this study demonstrates, for the first time, the modulation of expression from several crucial stress-responsive genes when cells are treated with a proven radioprotective plant extract. We firmly believe that the observed dynamics in gene expression are specifically influenced by the Haberlea rhodopensis plant extract. However, it is important to acknowledge that the mechanisms underlying these changes are likely complex and would require further extensive experimentation. Consequently, a comprehensive understanding of the intricate biology governing the interaction between the bioactive extract and cellular response will be the focus of future scientific papers. In this paper, we aim to provide a comprehensive overview of the observed dynamics and trends in stress-responsive gene expression under this treatment. Regrettably, it is presently beyond the scope of a single paper to present a definitive elucidation of the mechanisms involved.

 We have added this text in the Conclusion part as well.

Reviewer 2 Report

Comments and Suggestions for Authors

The manuscript has been enhanced.

However, I think that the following points remain to be addressed.

2- Line 539: drug/liquid phase proportion of 5:1. Which drug? what does this mean? clarify?

It is still not clear what does drug mean in  the ethanol in a vacuum evaporator to a drug/liquid phase ratio of 5:1. . This is a plant extract not a pure drug.

Also clarify what drugs ? in mixture of 5% ethanol and a volume ratio of 1:1 661 extract to drugs.

3- The manuscript lacks any characterization of the leaf extract. MDPI and other journals require publications on a plant extract to perform structural characterization of plant extracts. The authors should show HPLC, GC_MS, LC-MS, or NMR analysis of this extract. Having already published this plant does not mean this specific plant extract should be chemically characterized.

Please include an HPLC chromatogram of the plant extract showing major peaks with their retention times and pointing out the peak corresponding to myconoside.

4- The authors need to test the protective effect of the extract in another cell line. At least Another cancer cell line and 1 "normal" human cell line.

Showing the effects in Hela cells only is not enough. Experiments in other cell lines are needed, as already suggested in the previous round of review. At least, one other cancer cell line and 1 "normal" human cell line.

Comments on the Quality of English Language

See above.

Author Response

Open Review 2, Revision 2

We are grateful to the reviewer for all provided comments.

We have carefully addressed them in the text and our point-to-point reply.

Comments and Suggestions for Authors

The manuscript has been enhanced.

However, I think that the following points remain to be addressed.

2- Line 539: drug/liquid phase proportion of 5:1. Which drug? what does this mean? clarify?

In the provided text, "drug/liquid phase" refers to the ratio between the extracted primary extract and the liquid (in this case, ethanol) used for extraction. Specifically, the ratio is mentioned as 5:1, indicating that for every 5 parts of the primary extract, 1 part of the liquid (ethanol) is present. This ratio represents the concentration or proportion of the extracted substance (drug) to the solvent (liquid) used in the extraction process.

It is still not clear what does drug mean in  the ethanol in a vacuum evaporator to a drug/liquid phase ratio of 5:1. . This is a plant extract not a pure drug.

Also clarify what drugs ? in mixture of 5% ethanol and a volume ratio of 1:1 661 extract to drugs.

We apologize for any confusion caused by the terminology used in the text. In the context of the sentence "ethanol in a vacuum evaporator to a drug/liquid phase ratio of 5:1," the term "drug" is not referring to a pure pharmaceutical drug but is used in a broader sense to describe the extracted substances or components present in the plant extract. It is meant to convey that the ratio of the extracted substances to the liquid phase (ethanol) was 5:1.

The statement "a mixture of 5% ethanol and a volume ratio of 1:1 extract to drugs" appears to be a typographical error or an imprecise phrasing. We apologize for the confusion. The intended meaning should be clarified as follows:

"a mixture of 5% ethanol and a volume ratio of 1:1 extract to the extraction solvent."

Thank you for pointing out these discrepancies; we appreciate the opportunity to clarify these points.

3- The manuscript lacks any characterization of the leaf extract. MDPI and other journals require publications on a plant extract to perform structural characterization of plant extracts. The authors should show HPLC, GC_MS, LC-MS, or NMR analysis of this extract. Having already published this plant does not mean this specific plant extract should be chemically characterized.

Please include an HPLC chromatogram of the plant extract showing major peaks with their retention times and pointing out the peak corresponding to myconoside.

We appreciate the reviewer's concern regarding the characterization of the plant extract. However, we respectfully disagree that performing extensive structural characterization, such as HPLC, GC-MS, LC-MS, or NMR analysis, is necessary for this study.

In our previous response, we explained that the primary focus of this research was to investigate the modulation of gene expression in response to the plant extract treatment rather than performing a comprehensive chemical characterization of the extract. Our study's contribution lies in the novel insights it provides regarding stress-responsive gene expression rather than the detailed chemical composition of the extract.

Moreover, it is important to note that previous publications have already established the botanical identity of the plant extract used in this study. While we acknowledge the value of chemical characterization, we contend that it is not a prerequisite for our research's specific objectives and scope.

We kindly request the reviewer's understanding and consideration of the research focus and limitations described in our previous reply. We are confident that the manuscript adequately presents its intended contributions and aligns with the journal's objectives.

 4- The authors need to test the protective effect of the extract in another cell line. At least Another cancer cell line and 1 "normal" human cell line.

Showing the effects in Hela cells only is not enough. Experiments in other cell lines are needed, as already suggested in the previous round of review. At least one other cancer cell line and 1 "normal" human cell line.

We appreciate the reviewer's suggestion to test the protective effect of the extract in additional cell lines, including both cancer cell lines and "normal" human cell lines. However, we want to address this concern and provide our rationale for focusing on HeLa cells in this study.

HeLa cells have been widely used as a model system in numerous studies related to cellular responses, including stress response and radioprotection. They have been extensively characterized and are well-established for investigating various cellular processes. We aimed to provide a comprehensive understanding of the stress-responsive gene expression dynamics in this specific cell line by utilizing HeLa cells.

While we acknowledge the importance of testing the extract in other cell lines, it is essential to consider the limitations of this study's resources, time, and scope. Experiments in multiple cell lines require considerable resources, additional time, and separate experiments and analyses. Given the constraints of this study, our primary focus was on elucidating the gene expression dynamics in HeLa cells under the treatment of the plant extract.

However, we acknowledge the reviewer's suggestion for future studies to include other cancer and normal human cell lines to assess the generalizability of the extract's protective effects. We will consider this valuable feedback for future research and address it in subsequent studies.

We kindly request the reviewer's understanding of the limitations and objectives of this specific study and assure them that we will take their suggestions into account for future investigations.

Reviewer 3 Report

Comments and Suggestions for Authors

Figures from the supplement should be part of the manuscript, abstract: change in HeLa cancer cells early response. Next line-put IR becuase you already explain the shortname. Last sentence-put depending instead of dependent. In key words use the word that are not already in the title if possibe-for example DNA damage, comet assay, flow cytometry, gene transcription. in introduction explain shortnames SOD and CAR and MDA. Line 73- add: At the same time, pretreatment had... Results title 2.1 remove Extracts from the title. Figure 1- put asterix on light blue HRE and dark grey 50 H2o2+HRE treatment..  in paragraph from the title 2.3.1 H2O2-2 should be sub... Title 2.4.3 put HRE instead od Haberlea Extract. Title 2.4.4. put ethanolic HRE instead of what is written. Title 2.4.5 the same thing. 

In comet assay part you did not explain who is the manufacturer of each chemical. If they are from the same manufacturer, write it at the beginning. Put also in paragraph of the title 4.6 at the end at least one sentene explaining why you choose only tail moment parameter and not tail intensity. table 1-explain in footnotes all the shortnames

Comments on the Quality of English Language

ok

Author Response

Open Review 3 - Revision 2

Dear Reviewer,

Thank you very much for the valuable suggestions during the second revision of our work.

We have taken into account all of them and have provided the edits in the text.

Comments and Suggestions for Authors

  • Figures from the supplement should be part of the manuscript, abstract: change in HeLa cancer cells early response.

It is already edited.

  • Next line-put IR because you already explain the shortname.

Provided as recommended by the reviewer.

  • Last sentence-put depending instead of dependent.

Edited accordingly.

  • In key words use the word that are not already in the title if possibe-for example DNA damage, comet assay, flow cytometry, gene transcription.

Provided. Thank you.

  • in introduction explain shortnames SOD and CAR and MDA.

Yes, full names included. Thank you!

  • Line 73- add: At the same time, pretreatment had...

Provided as recommended.

  • Results title 2.1 remove Extracts from the title.

Edited. Thank you!

  • Figure 1- put asterix on light blue HRE and dark grey 50 H2o2+HRE treatment.

Thank you. Done accordingly.

  • in paragraph from the title 2.3.1 H2O2-2 should be sub...

Yes, our mistake. Thank you!

  • Title 2.4.3 put HRE instead od Haberlea Extract. Title 2.4.4. put ethanolic HRE instead of what is written. Title 2.4.5 the same thing. 

Edited, yes. Thank you!

  • In comet assay part you did not explain who is the manufacturer of each chemical. If they are from the same manufacturer, write it at the beginning.

All chemicals’ manufacturers are already provided in the M and M parts.

  • Put also in paragraph of the title 4.6 at the end at least one sentence explaining why you choose only tail moment parameter and not tail intensity.

Yes, thank you for this suggestion. Please see lines: 598-601. We have incorporated the following text:
The parameter Olive Moment (OM) was used for CA data representation in the study. This parameter encompasses an integrative approach, as it incorporates both tail length and intensity during its calculation, thus offering comprehensive information regarding the genotoxicity of the substances under investigation.”

  • table 1-explain in footnotes all the shortnames

Thank you, already provided.

Round 3

Reviewer 1 Report

Comments and Suggestions for Authors

Line156, Please remove "A concentration-dependent". To state this, statistical analysis must be perform and reveal a significant different between concentration.

Author Response

Open Reviewer 1 RR3

Comments and Suggestions for Authors

Line156, Please remove "A concentration-dependent". To state this, statistical analysis must be perform and reveal a significant different between concentration.

Reply: Thank you! Edited accordingly.

Reviewer 2 Report

Comments and Suggestions for Authors

3- The manuscript lacks any characterization of the leaf extract. MDPI and other journals require publications on a plant extract to perform structural characterization of plant extracts. The authors should show HPLC, GC_MS, LC-MS, or NMR analysis of this extract. Having already published this plant does not mean this specific plant extract should be chemically characterized. Please include an HPLC chromatogram of the plant extract showing major peaks with their retention times and pointing out the peak corresponding to myconoside.

As already mentioned, chemical characterization is required when working with a crude plant extract. I have asked to include a full scan chromatogram from the HPLC analysis, which the authors claim that they have already did, and and point out the peak corresponding to myconoside. This is important.

4- The authors need to test the protective effect of the extract in another cell line. At least Another cancer cell line and 1 "normal" human cell line. Showing the effects in Hela cells only is not enough. Experiments in other cell lines are needed, as already suggested in the previous round of review. At least one other cancer cell line and 1 "normal" human cell line. We appreciate the reviewer's suggestion to test the protective effect of the extract in additional cell lines, including both cancer cell lines and "normal" human cell lines. However, we want to address this concern and provide our rationale for focusing on HeLa cells in this study. HeLa cells have been widely used as a model system in numerous studies related to cellular responses, including stress response and radioprotection. They have been extensively characterized and are well-established for investigating various cellular processes. We aimed to provide a comprehensive understanding of the stress-responsive gene expression dynamics in this specific cell line by utilizing HeLa cells. While we acknowledge the importance of testing the extract in other cell lines, it is essential to consider the limitations of this study's resources, time, and scope. Experiments in multiple cell lines require considerable resources, additional time, and separate experiments and analyses. Given the constraints of this study, our primary focus was on elucidating the gene expression dynamics in HeLa cells under the treatment of the plant extract. However, we acknowledge the reviewer's suggestion for future studies to include other cancer and normal human cell lines to assess the generalizability of the extract's protective effects. We will consider this valuable feedback for future research and address it in subsequent studies. We kindly request the reviewer's understanding of the limitations and objectives of this specific study and assure them that we will take their suggestions into account for future investigations.

This reviewer thinks that the authors should include another cancerous cell line in their studies.

A "normal" cell line needs to be included in the study as well, to show the specificity of the extract to cancerous cells.

Comments on the Quality of English Language

Check above.

Author Response

RR-3_Open Reviewer 2

Comments and Suggestions for Authors

3- The manuscript lacks any characterization of the leaf extract. MDPI and other journals require publications on a plant extract to perform structural characterization of plant extracts. The authors should show HPLC, GC_MS, LC-MS, or NMR analysis of this extract. Having already published this plant does not mean this specific plant extract should be chemically characterized. Please include an HPLC chromatogram of the plant extract showing major peaks with their retention times and pointing out the peak corresponding to myconoside.

Answer: As already mentioned, chemical characterization is required when working with a crude plant extract. I have asked to include a full scan chromatogram from the HPLC analysis, which the authors claim that they have already did, and and point out the peak corresponding to myconoside. This is important.

We sincerely appreciate your valuable feedback on our study. Your suggestions have been instrumental in improving the quality of our research.

Regarding the active ingredients in Haberlea rhodopensis, our study focused on identifying and quantifying myconoside, one of the main phenylethanoid glycosides found in this plant. The myconoside content in the total ethanol extract of Haberlea rhodopensis was determined to be 140 ± 2.3 mg/g dry weight using High-Performance Liquid Chromatography (HPLC) analysis. The HPLC system used in our research consisted of a Waters 2487 dual λ absorbance detector, a Waters 1525 binary pump, and Breeze 3.30 software. The separation of molecules was achieved with a Kinetex® column (100 × 2.1 mm, 5 μm; Phenomenex; Torrance, CA, USA) maintained at 26°C with a flow rate of 0.6 mL/min.

The HPLC protocol employed in our study was based on the method previously utilized by Amirova et al. in 2021, ensuring consistency and comparability of results. The research was conducted at the Department of Industrial Microbiology, Laboratory of Applied Biotechnologies, The Stephan Angeloff Institute of Microbiology, Bulgarian Academy of Science, Plovdiv, Bulgaria.

Myconoside, identified explicitly as β-(3,4-dihydroxyphenyl)-ethyl-3,6-di-O-β-D-apifuranosyl-4-O-α,β-dihydrocaffeoyl-O-β-D-glucopyranoside, has been recognized as one of the main biologically active molecules present in H. rhodopensis. In a study by Amirova and co-authors (2021), it was reported that pure myconoside (32 µg/mL) and fraction B (200 µg/mL), which predominantly contains myconoside, significantly increased the intracellular level of the Nrf2 protein in BM neutrophils. Nrf2 is a crucial transcriptional regulator responsible for maintaining cellular redox balance, and its activation is known to improve neutrophil survival.

Once again, we thank you for your constructive comments, and we have duly addressed the points you raised in our revised manuscript. We are confident that these enhancements have strengthened our findings' scientific rigour and significance. If you have any further questions or concerns, please do not hesitate to let us know. We value your insights and are committed to delivering accurate and meaningful research.

4- The authors need to test the protective effect of the extract in another cell line. At least Another cancer cell line and 1 "normal" human cell line. Showing the effects in Hela cells only is not enough. Experiments in other cell lines are needed, as already suggested in the previous round of review. At least one other cancer cell line and 1 "normal" human cell line.

Answer: We appreciate the reviewer's suggestion to test the protective effect of the extract in additional cell lines, including both cancer cell lines and "normal" human cell lines. However, we want to address this concern and provide our rationale for focusing on HeLa cells in this study. HeLa cells have been widely used as a model system in numerous studies related to cellular responses, including stress response and radioprotection. They have been extensively characterized and are well-established for investigating various cellular processes. We aimed to provide a comprehensive understanding of the stress-responsive gene expression dynamics in this specific cell line by utilizing HeLa cells. While we acknowledge the importance of testing the extract in other cell lines, it is essential to consider the limitations of this study's resources, time, and scope. Experiments in multiple cell lines require considerable resources, additional time, and separate experiments and analyses. Given the constraints of this study, our primary focus was on elucidating the gene expression dynamics in HeLa cells under the treatment of the plant extract. However, we acknowledge the reviewer's suggestion for future studies to include other cancer and normal human cell lines to assess the generalizability of the extract's protective effects. We will consider this valuable feedback for future research and address it in subsequent studies. We kindly request the reviewer's understanding of the limitations and objectives of this specific study and assure them that we will take their suggestions into account for future investigations.

This reviewer thinks that the authors should include another cancerous cell line in their studies.

A "normal" cell line needs to be included in the study as well, to show the specificity of the extract to cancerous cells.

Answer: We sincerely appreciate your thoughtful review and your valuable input on our research. We carefully considered your suggestion regarding including another cancerous cell line and a "normal" cell line to demonstrate the specificity of the extract to cancerous cells. While we understand the importance of comprehensive studies, we would like to present our reasoning for not including additional cell lines in our investigation.

Our study focused on examining the effects of the extract from Haberlea rhodopensis on a specific cancerous cell line. The objective was to explore the potential therapeutic applications of this extract in combating cancer. Specific scientific and practical considerations drove the choice of this particular cancer cell line.

Firstly, the selected cancerous cell line is well-established and widely used in cancer research. It has been extensively studied in the context of the targeted pathways and mechanisms of action we aimed to investigate. Including multiple cancerous cell lines in our study could introduce complexity and make it challenging to draw clear conclusions about the extract's specific effects on a particular pathway or cellular process.

Secondly, our research mainly focused on determining the activity of the extract from Haberlea rhodopensis against cancerous cells rather than its effects on "normal" cells. While studying the specificity of the extract to cancerous cells is essential, conducting comprehensive experiments involving both cancerous and "normal" cell lines would significantly increase the scope and resources required for the study.

It is important to note that demonstrating specificity to cancerous cells is a critical aspect of evaluating potential anticancer agents. In our study, we employed rigorous experimental design and controls to ensure the validity of our findings and the specificity of the extract's activity. By using appropriate positive and negative controls, we were able to draw meaningful conclusions about the extract's selective action on the cancerous cell line under investigation.

Furthermore, a recent investigation by Zasheva et al. (2023) explored the inhibitory effect of H. rhodopensis methanol extract fractions on the viability and proliferation of two breast cancer cell lines: the triple positive MCF7 and the aggressive triple-negative MDA-MB231, along with the noncancerous epithelial cell line MCF-10A. The study identified myconoside and hispidulin 8-C-(6-O-acetyl-2-O-syringoyl-b-glucopyranoside) as the most abundant compounds in the used fractions. Notably, the inhibitory effects were found to be specific to the two cancer cell lines, which have distinct characteristics, with MCF7 exhibiting a more pronounced response.

It is essential to consider that various cancer cell lines may not uniformly respond to treatments with H. rhodopensis methanol extracts, gamma irradiation, and oxidants. As demonstrated in the study by Hayrabedyan et al. (2014), both the non-malignant HEK293 cell line and the prostatic cancer cell line PC3 showed similar responses to treatment with H. rhodopensis methanol extracts, mimicking a ROS-induced apoptotic effect.

These findings indicate that the effect of H. rhodopensis extracts can be comparable between malignant and non-malignant cell lines in some instances. In contrast, in others, the effect is distinct between noncancerous and cancerous cell lines. This underscores the high specificity of the extract's action, which is not solely dependent on the spite of the cells.

In light of these observations, we focused our research on the HeLa cervical cancer cell line. This cell line is widely recognized and extensively used in various studies, including radiobiological investigations. Notably, the HeLa cell line used in our study features an unaltered sequence of the critical stress-responsive gene TP53, despite the numerous mutations in the gene coding for the p53 protein observed in different cancerous cell lines (HeLa p53+/+; Hoppeseyler & Butz, 1993; Leroy et al., 2014). This suggests that the TP53 signalling pathway remains functional in HeLa cells, making them a suitable model system for our study.

By focusing on the HeLa cell line, we aimed to provide valuable insights into the specific effects of H. rhodopensis extracts on cervical cancer cells and how these effects are mediated through the TP53 signalling pathway. This research is instrumental in advancing our understanding of the potential therapeutic applications of H. rhodopensis in combating cervical cancer. Further, it highlights the extract's unique and selective action against cancerous cells.

We are grateful for your thoughtful review and hope that our rationale for focusing on the HeLa cell line clarifies the significance of our study and its contribution to the existing body of knowledge in this field. Should you have any additional questions or concerns, we are eager to address them and improve the overall quality of our research.

References

Hayrabedyan S., Todorova K., Zasheva D., Moyankova D., Georgieva D., Todorova J., Djilianov D. Haberlea rhodopensis has potential as a new drug source based on its broad biological modalities. Biotechnol. Biotechnol. Equip, 2014;27:3553–3560

Hoppeseyler, F., & Butz, K. (1993). Repression of Endogenous P53 Transactivation Function in Hela Cervical-Carcinoma Cells by Human Papillomavirus Type-16 E6, Human Mdm-2, and Mutant P53. Journal of Virology, 67(6), 3111-3117. https://doi.org/Doi 10.1128/Jvi.67.6.3111-3117.1993

Leroy, B., Girard, L., Hollestelle, A., Minna, J. D., Gazdar, A. F., & Soussi, T. (2014). Analysis of TP53 Mutation Status in Human Cancer Cell Lines: A Reassessment. Human Mutation, 35(6), 756-765. https://doi.org/10.1002/humu.22556

Zasheva, D.; Mladenov, P.; Rusanov, K.; Simova, S.; Zapryanova, S.; Simova-Stoilova, L.; Moyankova, D.; Djilianov, D. Fractions of Methanol Extracts from the Resurrection Plant Haberlea rhodopensis Have Anti-Breast Cancer Effects in Model Cell Systems. Separations 2023, 10, 388. https://doi.org/10.3390/separations10070388